# Statistical indicators of Arctic sea ice stability – prospects and limitations

S. Bathiany[1], B. van der Bolt[1], M. S. Williamson[2], T. M. Lenton[2], M. Scheffer[1], E. H. van Nes[1], D. Notz[3]

[1]Department of Environmental Sciences, Wageningen University, NL-6700 AA Wageningen, The Netherlands
[2]College of Life and Environmental Sciences, University of Exeter, United Kingdom
[3]Max-Planck-Institute for Meteorology, Bundesstrasse 53, 20146 Hamburg, Germany

*Correspondence to*: S. Bathiany (sebastian.bathiany@wur.nl)

**Abstract.** We examine the relationship between the mean and the variability of Arctic sea-ice coverage and volume in a large range of climates from globally ice-covered to globally ice-free conditions. Using a hierarchy of two column models and several comprehensive Earth System Models, we consolidate the results of earlier studies and show that mechanisms found in simple models also dominate the interannual variability of Arctic sea-ice in complex models. In contrast to predictions based on very idealised dynamical systems, we find a consistent and robust decrease of variance and autocorrelation of sea-ice volume before summer sea ice is lost. We attribute this to the fact that thinner ice can adjust more quickly to perturbations. Thereafter, the autocorrelation increases, mainly because it becomes dominated by the ocean water's large heat capacity when the ice-free season becomes longer. We show that these changes are robust to the nature and origin of climate variability in the models and do not depend on whether Arctic sea-ice loss occurs abruptly or irreversibly. We also show that our climate is changing too rapidly to detect reliable changes in autocorrelation of annual time series. Based on these results, the prospects of detecting statistical early warning signals before an abrupt sea-ice loss at a "tipping point" seem very limited. However, the robust relation between state and variability can be useful to build simple stochastic climate models, and to make inferences about past and future sea-ice variability from only short observations or reconstructions.

## 1 Introduction

The temporal evolution of Arctic sea ice in recent decades can be described by the superposition of a monotonous response to greenhouse gas forcing and internal climate variability (Notz and Marotzke, 2012). The latter determines the occurrence of extreme events, is key for the local perception of climate change (Hansen et al., 2012; Huntingford et al., 2013), and is closely linked to the stability of the mean state (Scheffer et al., 2009) and its sensitivity to forcing (Leith, 1975). In this contribution, we examine how the mean state and the variability of Arctic sea ice interact across a wide range of climate states. Our focus on Arctic sea ice is motivated by the fact that mean climate and variability are projected to show particularly large changes in the Arctic (Manabe and Wetherald, 1975; Huntingford et al., 2013), and processes linked to sea ice were put forward as major causes of these trends (Hall, 2004; Stouffer and Wetherald, 2007). However, the role of thermodynamic processes within Arctic sea ice itself, and its influence on the spectrum of the variability has not been discussed

in this context. Understanding the temporal evolution of variability in sea-ice area and volume also has practical consequences for example regarding the economic use of the Arctic. Moreover, understanding the relation between the mean climate and its variability will allow us to draw conclusions about the climate variability in the Earth's deep past, something that is difficult to reconstruct directly (White et al., 2010; Kemp et al., 2015), and that can help to build simple stochastic climate models.

Our focus is also driven by earlier speculations that Arctic sea-ice loss could reach a tipping point, i.e. a certain forcing where it would accelerate substantially (Lindsay and Zhang, 2005; Winton, 2006). Such a change is loosely referred to as 'abrupt' if the acceleration is due to mechanisms internal to the climate system (such as the positive ice-albedo feedback) whereas the forcing changes linearly over time (Rahmstorf, 2001; National Research Council, 2002). In the extreme case, the tipping point would correspond to a bifurcation point, a point of no return where sea ice is suddenly and irreversibly lost. While this "small ice-cap instability" occurred in simplified models (North, 1984; Thorndike, 1992; Eisenman and Wettlaufer, 2009; Abbot et al., 2011), more comprehensive models show a more gradual and reversible sea-ice loss in scenarios of the future (Armour et al., 2011; Tietsche et al., 2011; Boucher et al., 2012; Ridley et al., 2012; Li et al., 2013). Consequently, Wagner and Eisenman (2015a) recently showed in detail how resolving the seasonal cycle and latitudinal differences can eliminate bifurcations in sea-ice models, explaining why oversimplified models lead to wrong conclusions. Nonetheless, comprehensive models still differ in how abruptly Arctic sea ice area and volume can change (Bathiany et al., 2016). Given the large model uncertainties even in comprehensive models, it is worthwhile to investigate whether changes in certain aspects of the variability are specific to the existence of abrupt or even irreversible changes in the future. Observations might then provide an alternative source of information and indicate which model is most reliable in its prediction. Interestingly, when cooling the Earth instead of warming it, even comprehensive models show bifurcations, in agreement with simple models (Budyko, 1969; Sellers, 1969). For example, in a complex general-circulation model with current continental distribution and solar insolation, Marotzke and Botzet (2007) identified a globally ice-covered stable state analogous of the 'Snowball Earth' conditions in the Neoproterozoic (Pierrehumbert et al., 2011). Ferreira et al. (2011) and Rose et al. (2013) even found three stable states in a complex model with idealised ocean geometry. Climate variability plays an important role for the likelihood of transitions between such states, and for their reversibility (Lee and North, 1995), and thus needs to be considered to understand the evolution of climate in the Earth's deep past.

Furthermore, previous studies suggested that natural climate variability can be an indicator of climate stability and provide "early warning signals" of an approaching tipping point (Scheffer et al., 2009). The phenomenon of statistical stability indicators has long been known in mathematics (Wiesenfeld, 1985) and has later been applied to the problem of climate tipping points (Kleinen et al., 2003; Held and Kleinen, 2003). The theory applies to dynamical systems close to a stable fixed point that slowly destabilises over time. As the forces that restore a disturbed system towards equilibrium become weaker, the return rate to equilibrium becomes smaller, leading to an increasing relaxation time scale. Interestingly, this effect has been found in the simple deterministic climate model of Budyko (1969) when approaching the Snowball Earth bifurcation (Held and Suarez, 1974). In the presence of small perturbations in the form of a stochastic term added on the dynamic equation, it is often argued that 'slowing down' must cause an increase in autocorrelation and variance when approaching the tipping point

(Scheffer et al., 2009; Ditlevsen and Johnsen, 2010). In principle, this concept also applies to systems whose solution is not constant but periodic in time (Wiesenfeld and McNamara, 1986): By recording the state of a system at the same point in time during every period, a periodic solution can be transformed to a fixed point. However, the occurrence of statistical stability indicators relies on several assumptions such as the approximation of the system as one-dimensional (Held and Kleinen, 2004; Bathiany et al., 2013a), the assumption that the variability of the system results from small white noise external to the system (Dakos et al., 2012b), and that the system is close to its equilibrium solution. None of these assumptions is truly justified in the context of anthropogenic climate change. Even very simple stochastic models can deviate from the theory of statistical stability indicators due to the interactions of deterministic nonlinearities and noise (Dakos et al., 2012b, Bathiany et al., 2013a). Therefore, it is necessary to investigate if the approach can potentially yield meaningful results in the case of Arctic sea ice, and how the results depend on the model formulation. Moreover, even in cases when all assumptions hold, it is often not clear in practice how close a system needs to be to a bifurcation point for the theory to apply, and how slowly the destabilisation needs to occur to allow a significant detection of trends in variance or autocorrelation. In this study, we therefore also assess the practical applicability of statistical stability indicators to the problem of Arctic sea-ice loss by analysing simulations from models of very different complexity.

It follows already from previous studies that the total hemispheric ice area is not a suitable property to infer sea-ice stability. First, the distribution of continents determines where sea ice can occur and thus determines the variability of total sea-ice area (Goosse et al., 2009; Eisenman, 2010): While sea-ice area in the Arctic ocean is free to fluctuate, further south it is limited to the North Atlantic and North Pacific. The rest of the area is covered by continents which therefore 'mute' the variability of total sea-ice area (Eisenman, 2010). Second, when the latitude of the sea ice edge approaches the pole, there is less and less total area available in the (idealised) ice-covered circle (Goosse et al., 2009). Third, it has been noted that when sea ice becomes very thin, its open-water formation efficiency increases, meaning that small fluctuations in volume can lead to large fluctuations in area coverage (Holland et al., 2006; Goose et al., 2009; Notz, 2009). As all these effects result from geometrical constraints, they do not reflect the stability of the system in terms of its dynamical response to perturbations. We therefore focus on sea-ice thickness (or volume in a given area) in most of this study.

Regarding sea-ice volume and its relaxation time, previous studies rely on a small number of very idealised sea-ice models. For example, Merrifield et al. (2008) find increasing variance and an increasing relaxation time before an abrupt loss of summer sea ice in a simple model, apparently corroborating the classical concept of early warnings. However, the seasonal cycle is only parameterised crudely in their model, lumping processes of melting and freezing together in one equation. In a version of the simple column model by Eisenman and Wettlaufer (2009), Moon and Wettlaufer (2011, 2013) and Eisenman (2012) showed a relatively complex evolution of the system's timescale over a range of long-wave forcing with decreasing and increasing regimes due to a continuously changing balance of feedbacks. The most important positive feedback in this context is the ice-albedo feedback: Due to the ocean's low albedo compared to sea ice, ice loss and decreased surface albedo enhance each other. The most important negative feedback is the growth-thickness feedback: The thinner the ice becomes, the faster it can regrow due to an increased heat flux through the ice (Thorndike et al., 1975).

Moreover, the relatively large timescale of warming or cooling of the ocean's mixed layer becomes important once sea ice is not present during a substantial part of the year. Using a latitudinally explicit version of the model by Eisenman and Wettlaufer (2009), Wagner and Eisenman (2015b) therefore argue that the mixed-layer effect can raise false alarms of abrupt ice loss.

The above studies provide several scattered indications how the variability of sea ice may be linked to the mean climate. However, they are restricted to a small number of very simple sea-ice models that do not distinguish between ice area and volume and that are usually deterministic. As many nonlinear processes are disregarded that can affect the variability in non-intuitive ways, these studies allow only limited conclusions about the real world. In this study we aim for a systematic consolidating assessment by applying a hierarchy of sea-ice models that includes not only simple column models with one state variable but also comprehensive Earth system

models. Besides their different complexity, these models represent different scenarios of future sea-ice loss, namely a bifurcation-induced abrupt loss, an abrupt but reversible loss, and a gradual sea-ice loss. This allows us to demonstrate and explain a robust link between the mean, and the autocorrelation and variance of Arctic sea-ice volume that is not model-specific. In Section 2, we introduce the models we apply, explain their most important differences, and outline the setup of the simulations. In Sect. 3 we analyse the results of these

simulations, and we provide our conclusions in Sect. 4. Moreover, Appendix A provides additional information on the simplest of the models, and the changes we make to it in order to demonstrate different mechanisms.

## 2    Models and Methodology

### 2.1    Models

We apply different models in our study that all include a continuous annual cycle, the ice-albedo feedback and the growth-thickness feedback, and that are of very different complexity:

1. the box model by Eisenman (2012) with default parameters, here referred to as E12. The model is a slightly simplified version of the model by Eisenman and Wettlaufer (2009). It consists of a simple energy balance of the

ocean's mixed layer and describes the evolution of only one state variable, the enthalpy E. In the presence of sea ice, E is negative and proportional to the ice thickness, while during ice-free conditions, E is positive and proportional to the mixed-layer temperature. Hence, the model does not distinguish between ice-area coverage and ice volume because its ice-thickness distribution is a slab of ice with uniform thickness. The model equations are taken from Eisenman (2012) and reproduced in Appendix A. The effect of $CO_2$ is represented implicitly in

the surface net longwave balance $L_m$, which is our control parameter for this model. The model yields one stable solution with a perennial ice cover for present-day conditions, $L_m=1.25$ (as the model is non-dimensional, E and $L_m$ have no units). With decreasing $L_m$, the ice becomes thinner and the transition to a seasonal ice cover is gradual (Fig. 1a). In contrast, at $L_m \approx 0.925$, the remaining winter ice disappears abruptly due to a bifurcation in the system. Beyond this bifurcation point the only remaining stable cycle is ice-free during the whole year.

2. the box model by Eisenman (2007), referred to as E07. Like E12, it solves the energy balance of the mixed layer, taking solar radiation and atmospheric composition as boundary conditions. For the model equations including their derivation see Eisenman (2007). In contrast to the model by Eisenman (2012), several variables are explicitly modelled by ordinary differential equations: the ice area coverage A, the ice volume V, the surface temperature of the ice $T_i$, and the temperature of the mixed layer, $T_{ml}$. The ice has only a single thickness determined from h=V/A. The evolution of ice area is described by a parameterisation based on Hibler (1979). This allows one to explicitly distinguish sea-ice area from sea-ice volume. Atmospheric $CO_2$ is prescribed and given as a factor multiplied to the present-day concentration. Due to the fact that many processes have been intentionally neglected, the original model is rather insensitive to $CO_2$. To obtain a similar sensitivity than with the comprehensive model MPI-ESM we have added an additional flux of 16 $Wm^{-2}$ per $CO_2$ doubling to the downwelling long-wave radiation at the surface (eq. 30 in Eisenman (2007)). In similarity to E12, the model shows a gradual loss of summer sea ice, but an abrupt loss of winter sea ice under warming (Fig. 1b, c). The abrupt winter ice loss mainly results from the fact that the large ice area that forms each winter does not form anymore when the ocean no longer cools to the freezing temperature (Bathiany et al., 2016). In contrast to a bifurcation that results from a positive feedback, this abrupt change at a threshold is reversible. As the ice-albedo feedback is active also in E07, it produces a regime with multiple solutions around the critical $CO_2$ value. However, this regime is extremely small and thus not practically relevant.

3. the comprehensive Earth system model of the Max-Planck-Institute for Meteorology (MPI-ESM) (Giorghetta et al., 2013). In comparison to the two box models, the Earth system model MPI-ESM is much more complex. As a spatially explicit, comprehensive model, it describes a large number of processes considered relevant for the evolution of sea ice, including mechanical thickness redistribution and horizontal transport. Despite this huge process complexity, the description of the ice-thickness distribution is relatively simple (Notz et al., 2013): Similarly to E07, only one thickness class is used and the sea-ice area is calculated using the parameterisation by Hibler (1979). A further similarity with E07 is that an abrupt loss of winter sea-ice area occurs due to the simple representation of the model's ice-thickness distribution and the homogeneity of the Arctic Ocean (Bathiany et al., 2016). The abrupt ice loss is reversible (Li et al., 2013) and is caused by the same threshold effect as in E07 (Bathiany et al., 2016).

4. We also analyse eight additional comprehensive models from the Coupled Model Intercomparison Project 5 (CMIP5), using simulations of the historical period, the RCP8.5 scenario and its extension until the year 2300. The models are all the available models that lose their Arctic winter sea ice in these simulations. The level of complexity in these models is comparable to MPI-ESM, but some of them explicitly resolve several ice-thickness classes on the subgrid scale. While one of the models (CSIRO-Mk3-6-0) also produces an abrupt loss of winter sea-ice area like MPI-ESM, most models show a retreat of winter sea ice that is gradual (Hezel et al., 2014), though faster than the preceding summer sea-ice loss (Bathiany et al., 2016).

## 2.2    Methods

To investigate how the relaxation time, variance and autocorrelation in the models vary with $CO_2$, we perform three types of experiments that are reported in Sect. 3.1, 3.2 and 3.3 respectively, and whose technical details we address in these sections.

1. For the two column models E12 and E07 we perform numerical perturbation experiments where we run each model to its equilibrium annual cycle, then suddenly perturb it away from this reference solution by some small amount $x_0$, and measure the rate of return towards the reference solution. It follows from a linearisation of the system that the anomaly x decays exponentially over time t:

$$x = x_0 e^{-(t/\tau)} \tag{Eq. 1}$$

In systems with a time-independent solution, the relaxation time scale $\tau$ can essentially be obtained from only one specific state x at a time t by rearranging Eq. (1):

$$\tau = -\frac{t}{log\left(\left|\frac{x}{x_0}\right|\right)} \tag{Eq. 2}$$

Due to the permanent change in the balance of feedbacks during the annual cycle, the return rate in the models
we use depends on the time of year. To obtain a good estimate of the anomaly decay from year to year, we store t and x on December 21st in each year (the result is not sensitive to the choice of the date). We obtain the relaxation time $\tau$ from a linear regression of these annually resolved time series by regressing the numerator versus the denominator of Eq. (2).

2. For both column models, we perform stationary simulations and calculate the state variables' statistics. For
each of 50 different $CO_2$ levels we simulate 100,000 years under constant conditions. We then compute seasonal means for winter sea ice (averages over March to May), summer sea ice (averages over September to November), March and September. The definition of winter and summer sea ice captures the months of minimum and maximum sea ice volume in the models which lags the annual cycle of insolation. The time series of seasonal or monthly averages have annual resolution and are then used to calculate the autocorrelation (with a
lag of one year) and variance. With this approach we again focus on the effective relaxation time from year to year, and not the transient development of a perturbation within a year. In contrast to approach 1, the simulations involve stochastic terms that we add to the deterministic model equations. This involves choices on the place these terms are introduced in the equations (noise source), the magnitude of the noise (noise level) and its spectrum (color).

3. We analyse the trends of variance and autocorrelation in transient simulations of all models except E12 using a sliding window approach. As in the case of stationary simulations, all time series here are seasonal means, hence the time series have annual resolution. In particular, we analyse a simulation from MPI-ESM where $CO_2$ increases linearly until it has quadrupled after 2,000 years. This simulation has been performed and reported by Li et al. (2013). We perform similar simulations with E07, using different experiment lengths and 1,000
realisations per experiment. We also apply this method to the combined historical and RCP8.5 scenario

simulations from CMIP5. Such transient simulations with continuously increasing $CO_2$ concentration more closely describe the ongoing change of the real climate system than the idealised, stationary experiments described before.

## 3    Results

### 3.1    Deterministic perturbation experiments

We begin by analysing how the response time $\tau$ of the model by Eisenman (2012) depends on the surface long-wave balance $L_m$ (Fig. 2a). To this end, we perturb the state variable E by +0.005 and measure the decay rate over two years. The decay rate is thus determined from three points (start time, end of year one, and end of year two). Using more years leads to the same results but fails in cases when the system is very stable because the anomaly then becomes too small to be measured after only a few years. In agreement with Eisenman (2012) and Moon and Wettlaufer (2011), the response time shows a characteristic double-peak when increasing $CO_2$ (decreasing $L_m$). The first peak occurs at $L_m \approx 1$ where the summer ice is lost and the ice-albedo feedback is substantially strengthened due to the exposed open water during a growing period during the year. The second peak occurs at the bifurcation point $L_m \approx 0.93$ where the winter sea ice vanishes. To this extent, the system is in agreement with dynamical-systems theory that predicts a slowing down as a result of increasing positive feedbacks.

To show that both peaks are indeed caused by the ice-albedo feedback, we perform additional experiments where this feedback is switched off. Following Eisenman (2012), we do this by setting the albedo difference between ice and water to 0. Appendix A explains the changes we make to the model equations in order to switch off certain mechanisms. Fig. 2b shows the relaxation time $\tau$ for the model without ice-albedo feedback but no other changes. Obviously, the range of $L_m$ over which a complete ice loss occurs is much larger due to the removal of a positive feedback. The most striking change in the evolution of $\tau$ is that the bifurcation as well as the double peak in the relaxation time have disappeared. The role of the ice-albedo feedback in E12 has also been analysed analytically by Moon and Wettlaufer (2011) who obtained the same result.

Another striking feature in Fig. 2a and Fig. 2b is the large regime of decreasing $\tau$ from preindustrial conditions up to shortly before the loss of summer sea ice. This decline results from the fact that the heat conduction through the ice becomes more efficient with decreasing thickness. This is important during freezing conditions when the heat from the ocean has to diffuse through the ice before it can be radiated away from the ice's surface. Therefore, thin ice grows faster than thick ice, and the thinner the sea ice becomes, the more rapidly it can adjust to perturbations. Fig. 2c documents the validity of this interpretation: In addition to switching off the ice-albedo feedback, we also remove the growth-thickness feedback from the equations. To still obtain a stable system, the removed stabilising feedback is replaced by the negative Planck feedback that is also active in the ice-free season in the default model (see Appendix A). As a result, the relaxation time is constant in the regimes of perennial ice cover or open ocean. The fact that the response time decreases with ice thickness has implications for the transition to a Snowball Earth state: Cooling the climate towards such a state will result in an increasing

autocorrelation and variance, as a spatially explicit version of E12 now also shows (Wagner and Eisenman, 2015b). However, we stress that this effect is not a good example of successful "early warning signals": As the growth-thickness relationship is independent of the ice-albedo feedback and the existence of a bifurcation, variance and autocorrelation would also increase in absence of a catastrophic transition to a Snowball Earth state. Of course, knowing the variance of a system is still useful to estimate the probability of a transition to an alternative state if it is already known to exist.

A third alteration to the model reveals the reason for the increase of $\tau$ in the regime of seasonal sea ice (Fig. 2d). The difference to the version in Fig. 2b is that we halve the effective heat capacity of the ocean's mixed layer (e.g. representing a more shallow mixed layer). Obviously, this reduces the relaxation time for the ice-free system, because the model then simply consists of a well-mixed box of water whose heating or cooling rate is proportional to its mass. The warmer the climate becomes, the longer is the ice-free season, and the more does the system's effective timescale approach the timescale of an ice-free ocean. As this timescale is longer than the one of the thin sea ice, a "slowing down" occurs. Therefore, this increase in relaxation time is not related to any bifurcation approaching (there is none in Fig. 2b-d), or in fact to any positive feedback. This specific result has also been obtained in the latitudinally resolved version of E12 by Wagner and Eisenman (2015b).

In the following, we go a step further and show that the above results also hold in more complex models. We begin with the second column model, E07. Due to the four state variables in this model, it has to be decided how to perturb the system in the numerical perturbation experiments. In principle, a system responds differently depending on which state variable is perturbed. While the water's large specific heat capacity and latent heat of fusion determine the long-term slow response of the system, perturbations of the radiative fluxes decay very quickly. Our numerical perturbation experiment for E07 consists in a perturbation of $T_{ml}$ by +0.2 K. For the determination of the relaxation time via regression, we use years 2 and 3 after the perturbation is applied, ensuring that anomalies of the fast modes have already decayed after the first year.

Interestingly, for the loss of summer sea ice E07 displays an evolution of $\tau$ that matches the results from E12 with fixed albedo (Fig. 2b): A regime of decreasing $\tau$ during the loss of summer sea ice is followed by a regime of slightly increasing $\tau$ after the complete summer sea-ice loss (Fig. 3a). For winter sea ice, in contrast, results with E07 match the evolution of E12 with interactive albedo, with a narrow peak in relaxation just before the loss of winter sea ice. This peak disappears when the albedo feedback is disabled in E07, while the response time for the loss of summer sea ice remains largely unchanged (Fig. 3b). This demonstrates that the ice-albedo feedback is of secondary importance for the evolution and stability of summer sea ice in E07 (except at the very point of abrupt winter ice loss). It is important to note that both models roughly show the same evolution of the relaxation time (decreasing during summer ice loss, increasing thereafter), regardless of whether there is a bifurcation or any abrupt change approaching or not.

### 3.2 Stationary stochastic simulations

In natural systems, the relaxation time usually cannot be measured or calculated as directly as in models. However, one can hope to measure the system's response to natural external perturbations indirectly in form of its variance and autocorrelation. We therefore investigate in stochastic versions of the two column models whether these indicators reflect the changes in timescale. In each experiment, we introduce noise in one of three terms of the equations: To mimic variability in the ocean heat flux ($\sigma_{OHF}$), we added a Gaussian white noise term to Eq. A1 (Appendix A). To introduce noise to the radiative fluxes, we added the noise term on the radiative balance A (Eq. A2) to perturb the long-wave balance ($\sigma_{LW}$), or on S ($S = 1 - S_a \cos 2\pi t$) in Eq. (A2) to perturb the short-wave balance ($\sigma_{SW}$). We also distinguish small and large noise, as well as white and red noise. In the case of small noise, we choose the noise level in a way that the total variance of E is in the order of $10^{-9}$, i.e. much smaller that the amplitude of an annual cycle. In the case of large noise, we adjust the noise level such that the system's stochastic variability is roughly one order of magnitude smaller than the amplitude of the annual cycle, in similarity to the situation in the real world. In case of red noise, we model the external perturbations as an autoregressive process of order one (AR(1) process) with a decorrelation time of 180 days in case of mixed layer energy and 10 days for atmospheric radiation.

Fig. 4 shows results for large red noise for all three noise sources. Interestingly, the specific choices for the noise terms hardly affect the results. When introducing small noise to the equations, the evolution of variance and autocorrelation closely follow the results we obtained from the perturbation experiments (Fig. 2a), independent of the noise type. Due to the low temperatures and the large growth-rate of thin ice, the ice coverage A is always close to 1 in winter, and has very small variance regardless of the variability of other variables. In contrast to Fig. 2a, the second peak produced by the ice-albedo feedback is not as pronounced in Fig. 4. This partly results from the lower resolution of the figure (associated with the much larger computational demand), but mostly due to the fact that the natural variability causes the system to cross the tipping point before the deterministic bifurcation point is reached. However, even in case of large red noise, the results are qualitatively similar to Fig. 2a as long as the noise is still small enough to not destroy the whole bifurcation structure of the system. The reason is that the time scales of the variability are still smaller than the typical response time of the ice-mixed layer system. In this regard, the model still sees the imposed noise as white, and the autocorrelation we find is determined by the system's time scale and not the time scale of the red noise. This explains the invariance of the results to the noise type.

In the stochastic experiments with E07, we introduce the stochastic terms in the same way as in the case of E12, and we find the same robustness to noise source, level and colour (Fig. 5). Naturally, the variance of the summer sea-ice area shows a distinct peak before the thickness approaches 0 because A becomes very sensitive to small perturbations in V (first row in Fig. 5). The peak occurs when fluctuations in A are least affected by the lower and upper limits, A=0 and A=1, and variance decreases thereafter because a larger fraction of the summer is already ice free, thus reducing the possible total variability. As sea ice disappears first in September, the variability peak occurs first in this month. The increased open-water formation efficiency before total ice loss has been reported in previous studies (Holland et al., 2006; Goosse et al., 2009) and is most evident during the

gradual process of summer ice loss (around CO2≈1.9). The phenomenon is confined to a very narrow parameter regime for the abrupt winter ice loss around CO2≈3.4 because the rapid growth of new ice each winter tends to keep A close to 1 until shortly before total ice loss (Bathiany et al., 2016).

The evolution of the volume's autocorrelation (last row in Fig. 5) closely follows the timescale obtained from the perturbation experiment. In principle, this is also true for the variance of ice volume fluctuations although it does not show a clear increase after summer ice loss (third row in Fig. 5). Interestingly, the autocorrelation of ice area (second row in Fig. 5) does not show the same evolution as the autocorrelation of ice volume in the regime of perennial ice. Such non-intuitive behaviour can occur as a result of the noise propagation through the nonlinear system and due to the permanent changes of feedbacks in different times of the year (Moon and Wettlaufer, 2013). Ice-area anomalies tend to have a shorter time scale than volume adjustments, especially due to the rapid growth of new ice that can quickly produce a large area increase with only small volume changes. Therefore, the autocorrelation of ice area is usually smaller than that of ice volume (it should be noted that the autocorrelation of the area fraction of winter sea ice has little practical relevance for most $CO_2$ levels because A is always very close to 1, as is reflected by its very small variance). As it is the slowest mode that dominates the relaxation time of the full system, the autocorrelation of ice volume corresponds well to the time scales we measured in the perturbation experiments (Fig. 2).

### 3.3    Transient stochastic simulations

We now analyse transient simulations with E07 and compare them to the most comprehensive model, MPI-ESM. We focus on the evolution of ice volume and its statistics. Each experiment starts from pre-industrial $CO_2$, which is then quadrupled over 2000 years. After approx. 1550 years, an abrupt loss of Arctic winter sea ice occurs in both models. As the description of the ice-thickness distribution is similar in E07 and MPI-ESM, the abrupt winter sea-ice loss probably results from the same threshold mechanism (Bathiany et al., 2016). This is corroborated by the fact that the abrupt loss is reversible in MPI-ESM (Li et al., 2013).

We use red noise that is added to the ocean heat transport and that has a noise level which produces a similar variability in ice volume as MPI-ESM at individual grid cells. To obtain the evolution of variance and autocorrelation of the ice volume in both models we apply the 'early warnings' R package described in Dakos et al. (2012a), which performs an analysis often applied to transient time series (Lenton, 2011). The method consists in a running window of 300 years that slides from the beginning of the time series to the point just before the ice loss. In the case of summer sea ice, this final point is reached after 800 years, in the case of winter ice loss after 1550 years. As in the case of the stationary simulations, each time series consists of annually spaced seasonal means. Within the running window, fluctuations on long timescales are removed by smoothing the time series with a Gaussian kernel of interactive bandwidth and subtracting this smoothed version from the original time series. For the residuals, variance and autocorrelation are calculated within the window. As the window moves along the time series, we obtain an evolution of variance and autocorrelation (being shorter than the original time series by the window length).

The results with E07 are similar to the stationary experiments in the previous section (Fig. 6). As the simulations are much shorter than the stationary experiments, the results are much noisier. However, the decrease in variance (Fig. 6 d-f) and the decline and subsequent increase in autocorrelation of V (Fig. 6 g-i) are still clearly visible. As MPI-ESM is a spatially explicit model, one has to choose a specific region. We analysed six different single grid cells in the Arctic ocean and obtain a qualitatively similar evolution of statistics; Fig. 6 b, e, and h show results for a grid cell located at approx. 102 W and 86.5 N. Fig. 6 c, f, and i show results for the total ice volume north of 75 N. Thus, the behaviour at individual grid cells carries over to the regional scale. The results from MPI-ESM are also in good agreement with the results from E07 – the inclusion of spatial differences and processes like advection and mechanical redistribution of sea ice apparently has not changed the behaviour of sea ice variability. We therefore argue that E07 is an appropriate model to explain the behaviour in MPI-ESM and it is probable that the same processes are behind the evolution of the statistics. This finding is corroborated by the fact that the abrupt loss of winter sea ice is also due to the same reason in both models (Bathiany et al., 2016).

As Fig. 6 only presents a single realisation from both models the question arises how long a time series needs to be in order to observe significant trends. We therefore conducted four different experiments with E07 where the quadrupling of $CO_2$ occurs over 100, 200, 500 and 2000 years, respectively. For each experiment we perform 1,000 realisations and calculate the trends in variance and autocorrelation in each realisation. These trends are given as Kendall Tau values that express how monotonically a property changes. A time series with only positive (negative) changes from one point to the next has a Kendall Tau of 1 (-1), a time series with an equal number of increases and decreases has a Kendall Tau of 0. Fig. 7 shows the distribution of Kendall Tau values for the trends in variance and autocorrelation of winter sea ice. Sea ice loss occurs at slightly different times in the different realisations. Winter sea-ice loss typically occurs after 4/5 of the experiment length. In each realisation, the sliding window in which variance and autocorrelation are measured therefore stops 5 years before zero ice volume occurs for the first time in September (Fig. 7a,b) or March (Fig. 7c). Increasing the window length improves the results, but the window length is of course limited by the length of the time series. We therefore chose a constant relative window length of 3/20 of the experiment length. The results somewhat depend on the details of this analysis and the system under consideration. However, Fig. 7 illustrates that several hundred to thousand years are required to obtain robust trends. While these results support our interpretation that the 2,000-year experiments in Fig. 6 are meaningful, simulations with more plausible scenarios cannot be expected to yield robust results. In general, variance is better constrained than autocorrelation (Ditlevsen and Johnsen, 2010). Therefore, one can expect to see a decrease in variance of sea-ice volume but no consistent changes in autocorrelation in simulations where sea-ice loss occurs within less than 200 years, a typical experiment length for projections of anthropogenic climate change.

To test this prediction, we finally analyse CMIP5 simulations from MPI-ESM and eight other comprehensive climate models. For this analysis we combine the historical simulation, the RCP8.5 simulation, and the extended RCP8.5 simulation that ends in the year 2300. In this scenario, atmospheric $CO_2$ shows an accelerated increase until the year 2100, when a radiative forcing of approx. 8.5 W/m$^2$ is reached. Thereafter, the $CO_2$ concentration stabilises at almost 2000 ppm (Meinshausen et al., 2011), yielding the largest warming of all CMIP5 simulations. The extended simulations until 2300 were performed with nine models (Hezel et al., 2014). Here we analyse all

models where Arctic sea-ice area falls below one million square kilometres in the full RCP8.5 scenario, no matter when this event occurs. Two of the models analysed in Hezel et al. (2014) do not lose their winter sea ice by 2300, while two other models not analysed by Hezel et al. (2014) have lost their winter sea ice already by 2100 (the nine models we analyse are therefore not identical to the nine models in Hezel et al., 2014). For the analysis of the CMIP5 simulations we use the same sliding window approach as explained above, using a window length of 30 years. The results confirm our findings from above: Sea ice variance decreases in most models (especially those with a large pre-industrial variability). The decrease in variance occurs not only in the whole Arctic but also at individual grid cells and is thus likely to result from the increasing growth-thickness feedback discussed in Sect. 3.1. Autocorrelation shows no convincing signal compared to Fig. 6 which is not surprising given the short time series (Fig. 8), though a hint of a decrease in autocorrelation seems to be visible. As we have shown in the previous sections, the trends in variance and autocorrelation that occur in sufficiently long simulations are not specific to the mechanism of ice loss. Fig. 7-8 illustrate yet another limitation to the applicability of early warning signals: Even if there was any information in these trends, it would be impossible to detect it in a single realisation with the current rate of global warming. As we analyse seasonal means with a time step of one year from one data point to the next, a higher temporal resolution may provide an improvement (e.g. Williamson et al., 2016). This would require one to remove the annual cycle from the time series before the statistical analysis of the anomalies. However, as the annual cycle changes considerably when sea ice is lost, and as other relevant mechanisms might come into play on short time scales, we leave this challenge to future studies.

## 4    Conclusions

Using a hierarchy of models, we have demonstrated a robust link between the mean state and the variability of sea ice. This link concerns all climate states between a perennial ice cover and a perennial open Arctic Ocean. While the relaxation time of Arctic sea ice tends to decrease before summer ice loss, it increases before winter ice loss in all models. In time series of sea-ice volume these trends carry over to autocorrelation and, to some extent, variance. The decreasing response time during summer sea-ice loss is caused by the more efficient heat conduction through the thinning ice. The increasing response time during winter sea-ice loss is mainly caused by the long response time of the ocean which becomes more influential as the ice retreats. We found that these results do not depend on whether Arctic sea-ice loss occurs abruptly or even irreversibly in a model. At first sight, this may appear to be in conflict with the generic concept of slowing down. In principle, however, the concept does apply to the case of sea-ice loss: Just before the bifurcation at the point of winter sea-ice loss occurs in E12 or E07, a sharp peak emerges (Fig. 2a, Fig. 3a). The peaks are more pronounced when the ice-albedo feedback is important like in E12, where $\tau$ even peaks during summer ice loss, and less pronounced before the winter ice loss in E07 which is mainly due to a threshold mechanism.

The practical problem is that these bifurcation-induced peaks occur in such a narrow parameter regime that it will be impossible to detect them before an abrupt change in transient time series. The general trends in transient time series will therefore be independent of the mechanism or even the existence of an abrupt change. In order to

infer information on the system from its variability, these trends would need to be more specific to certain mechanisms. In models of low or intermediate complexity however, it may well be possible to investigate the mechanism of an abrupt change diagnostically by creating long stationary time series for carefully selected forcing conditions (Bathiany et al., 2013a,b). It is therefore useful to see that our results are robust to the source

of the noise, its spectrum and its magnitude. We have also shown that long simulations are necessary to obtain robust results, typically more than 1000 years (also see Ditlevsen and Johnsen, 2010). This is also the reason why it will be difficult to see consistent trends in observations. Livina and Lenton (2013) found a recent increase in autocorrelation for summer sea-ice area from satellite observations when corrected for the continental distribution, but no other clear signals due to the rather short record. Similarly, Williamson et al. (2016) found no

change in the phase lag of Arctic sea-ice area relative to the annual cycle of insolation, indicating no change in relaxation time. The column models we applied suggest that ice volume (or thickness in the column models), captures the system's relaxation time better than area fraction. Unfortunately, ice thickness and volume are much more difficult to observe than sea-ice area. We conclude that if sea ice was approaching a tipping point, observations of sea-ice variability would not help to predict it.

The comprehensive model we analysed in most detail, MPI-ESM, likely exaggerates how rapidly the final bit of winter sea-ice volume disappears (e.g. as seen in the top right panel of Fig. 8). This abrupt volume loss is probably related to the ice-growth parameterisation, which attributes a single thickness to all newly formed ice in a grid cell (Bathiany et al., 2016). Although the abrupt event itself is not part of our time series analysis above, it points to potential limitations of the applied model and one may ask how models with several ice-thickness

classes would behave. It is reassuring in this regard that eight other models agree with MPI-ESM in their decrease of the sea-ice volume's variance, although time series were too short to show clear trends in autocorrelation. Moreover, the mechanistic insight obtained with the simpler models suggests that these model agreements are no coincidence because they can be explained from fundamental physical processes. Both the fast adjustment of thin ice and the slow response of the mixed-layer ocean are represented in all the models and

would also not change in even more complex models. For example, in models with many ice-thickness classes, the variability of the total ice volume in a grid cell is the result of the variability of all thickness classes. The trends in variance and autocorrelation would have the same sign for each thickness class because the thickness-growth relationship is monotonous (Thorndike et al., 1975). Even the precise realisation of the weather-induced variability would be identical because all thickness classes within a grid cell are coupled to the same ocean and

atmosphere grid cell. Hence, the level of sophistication in the representation of the subgrid-scale ice-thickness distribution is not relevant for our results. Furthermore, it has been shown in Bathiany et al. (2016) that radiative feedbacks and mechanical redistribution mechanisms are unimportant for the abruptness of sea-ice loss in MPI-ESM, which is instead determined by thermodynamic processes. It is therefore plausible that the same processes also determine the variability of sea ice before the final ice loss occurs.

Interestingly, our result that the relaxation time is unrelated to the existence of a tipping point has analogies in many other systems whose effective 'mass' changes over time. For instance, the effective heat capacity of the ocean increases with the mixed layer depth, which can cause an increase in autocorrelation although the system does not destabilise (Boulton and Lenton, 2015). Moreover, the relaxation time scale may depend on the

direction of perturbations, just like sea-ice melting and freezing is determined by different processes. An example for such asymmetry is vegetation dynamics (Bathiany et al., 2012): While vegetation can die back or burn within days or months, its regrowth can take many decades. Such restrictions and the fact that the statistics of sea ice in the models are closely linked to its mean state may make the prospect of 'early warnings' for accelerated sea ice loss appear rather limited. However, this also provides opportunities. First, the physical mechanisms behind the phenomena we have described are relevant for paleoclimate problems such as the role of sea-ice variability in the Eocene (White et al., 2010), or the transitions into and out of a "Snowball Earth" state (Pierrehumbert et al., 2011). For example, our results would allow one to formulate a stochastic parameterisation of sea-ice variability for simple climate models that is valid in all climates. Second, due to the crucial role of sea ice in the Arctic climate, an improved understanding of sea-ice variability will contribute to understand the future evolution of Arctic climate variability in general (Stouffer and Wetherald, 2007; Huntingford et al., 2013). In particular, the strict relation between the mean state of sea ice and its variability suggests the possibility to infer the system's total variability from relatively short observational time series, and to estimate the typical magnitude and longevity of climate anomalies in the future. This knowledge will be important for ecosystems and economical activities in the high latitude oceans.

## Appendix A. Description of Eisenman (2012) model and feedback suppression method

Here, we describe the model by Eisenman (2012), denoted E12 in the main text, and the changes made to separate different effects.

The dynamic equation of the model is

$$\frac{dE}{dt} = A - BT + F_B \tag{A1}$$

with t for time and E for enthalpy. In the presence of sea ice, E is negative and proportional to the ice's thickness, while during ice-free conditions, E is positive and proportional to the mixed-layer temperature.

Term A in Eq. A1 describes the temperature-independent terms of the radiative balance

$$A = \left(1 + \Delta_\alpha \tanh\left(\frac{E}{h_\alpha}\right)\right)(1 - S_a \cos 2\pi t) - L_m - L_a \cos 2\pi(t - \Phi) \tag{A2}$$

with $L_m$ is the annual mean long-wave radiation balance at the surface, the control parameter we vary in our experiments to represent a change in atmospheric $CO_2$.

T represents the surface temperature of the ice-ocean system and is calculated from

$$T = \begin{cases} E, & E \geq 0 \quad \text{[open ocean]} \\ 0, & E < 0, A > 0 \text{ [melting surface]} \\ \frac{A}{B}\left(1 - \frac{\zeta}{E}\right)^{-1}, & E < 0, A < 0 \text{ [frozen surface]} \end{cases} \tag{A3}$$

These equations correspond to Eq. 9-11 in Eisenman (2012) with the exception that we have omitted the tilde above all variables that denotes them as non-dimensional variables. All parameter values are listed in his Tab. 1. For a derivation of these equations and an explanation of all parameters see Eisenman (2012) and Eisenman and Wettlaufer (2009).

To switch off the ice-albedo feedback, we set $\Delta\alpha$ to zero. To switch off the growth-thickness feedback in addition, the temperature equation is replaced by

$$T = \begin{cases} E, & E \geq 0 \text{ [open ocean]} \\ 3E, & E < 0 \text{ [sea ice]} \end{cases} \tag{A4}$$

This way, the stabilising growth-thickness feedback is replaced by the stabilising Planck feedback, the same that also operates under ice-free conditions. The factor 3 in the presence of sea ice is arbitrary and was introduced merely to distinguish the regime with and without sea ice in Fig. 2c.

Alternatively, to reduce the heat capacity of the mixed layer by a factor 2 we exchange the temperature equation by

$$T = \begin{cases} 2E, & E \geq 0 \qquad \text{[open ocean]} \\ 0, & E < 0, A > 0 \text{ [melting surface]} \\ \frac{A}{B}\left(1 - \frac{\zeta}{E}\right)^{-1}, & E < 0, A < 0 \text{ [frozen surface]} \end{cases} \tag{A5}$$

As the equations are dimensionless, the mixed-layer heat capacity C does not explicitly appear in the model equations. In case of open water, E incorporates the inverse of C, which is why halving C corresponds to
doubling E in the open water case of the above equation (see Eisenman (2012) for details on the model derivation).

**Acknowledgements**

This work was carried out under the programme of the Netherlands Earth System Science Centre (NESSC), financially supported by the Ministry of Education, Culture and Science (OCW). We also acknowledge the World Climate Research Programme's Working Group on Coupled Modelling, which is responsible for CMIP, and we thank the climate modeling groups for producing and making available their model output. We thank Vasilis Dakos for helping to apply his early warnings R package and Chao Li for making available the MPI-ESM
model output. S.B. gratefully acknowledges Arie Staal for his fruitful and revealing approaches to savour scientific achievements. We are also indebted to Till Wagner and Ian Eisenman for their valuable comments and

their very amiable and cooperative spirit. Finally, we acknowledge two anonymous reviewers who helped us to improve the manuscript.

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

30

**Figures**

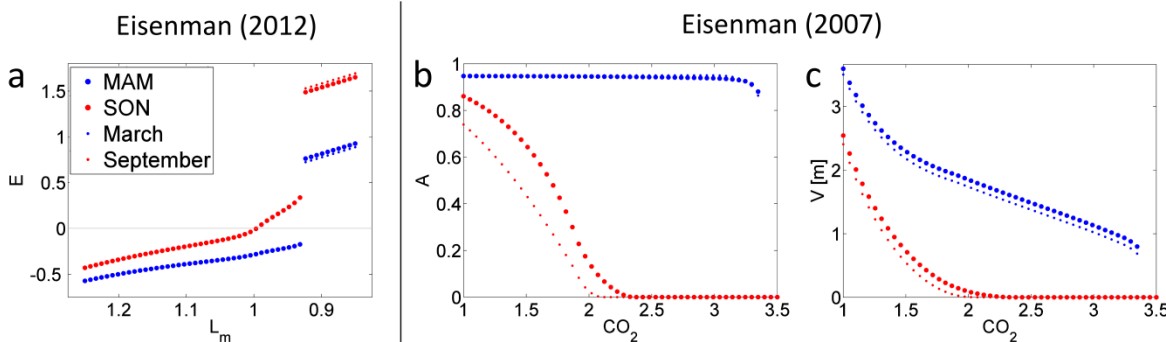

Figure 1. Response of the two column models to warming. a) Enthalpy versus surface longwave balance $L_m$ in E12. The horizontal line demarcates between positive E (open water) and negative E (ice covered ocean). b) Ice-area fraction and c) ice volume (given as an equivalent thickness) versus $CO_2$ (given as multiples of the pre-industrial value) in E07. Each dot represents a time mean over the season indicated in the legend of subfigure (a).

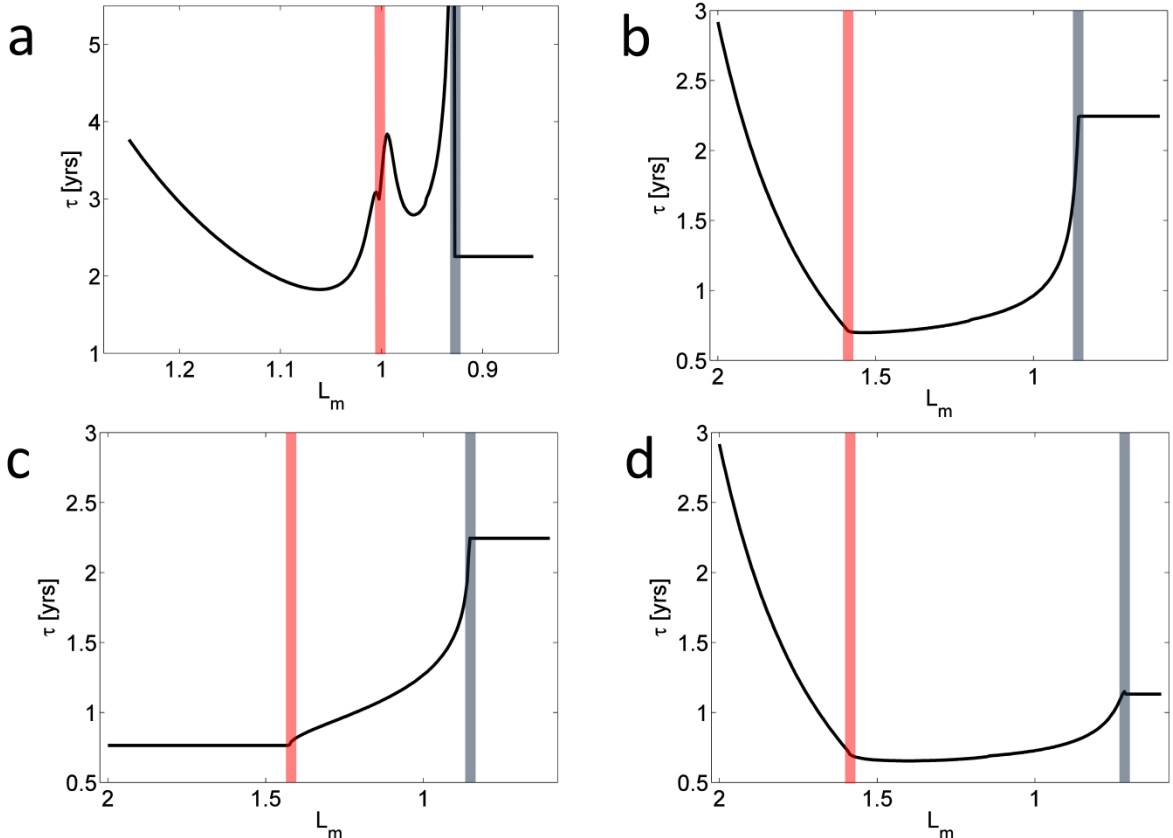

Figure 2. Relaxation time scale in Eisenman's (2012) box model for different combinations of mechanisms. a) Original model, b) original model but with disabled ice-albedo feedback; c) like (b) but without growth-thickness feedback; d) like (b) but with only half the default ocean heat capacity. The vertical shaded lines indicate the values of $L_m$ where the annual minimum (red) and maximum (blue) ice volume reaches zero.

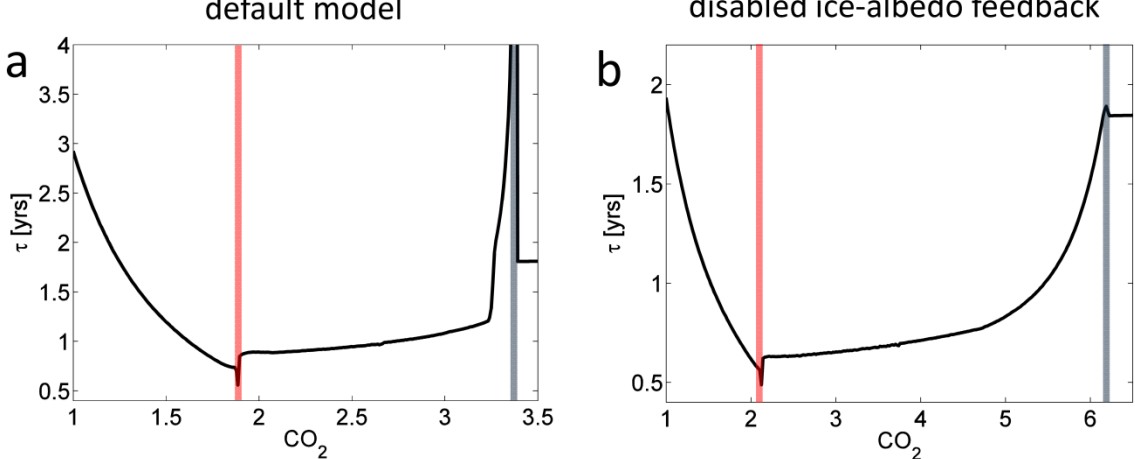

Figure 3. Relaxation time scale in Eisenman's (2007) box model for a) the original model, b) with disabled ice-albedo feedback. The vertical shaded lines indicate the values of $CO_2$ where the annual minimum (red) and maximum (blue) ice volume reaches zero.

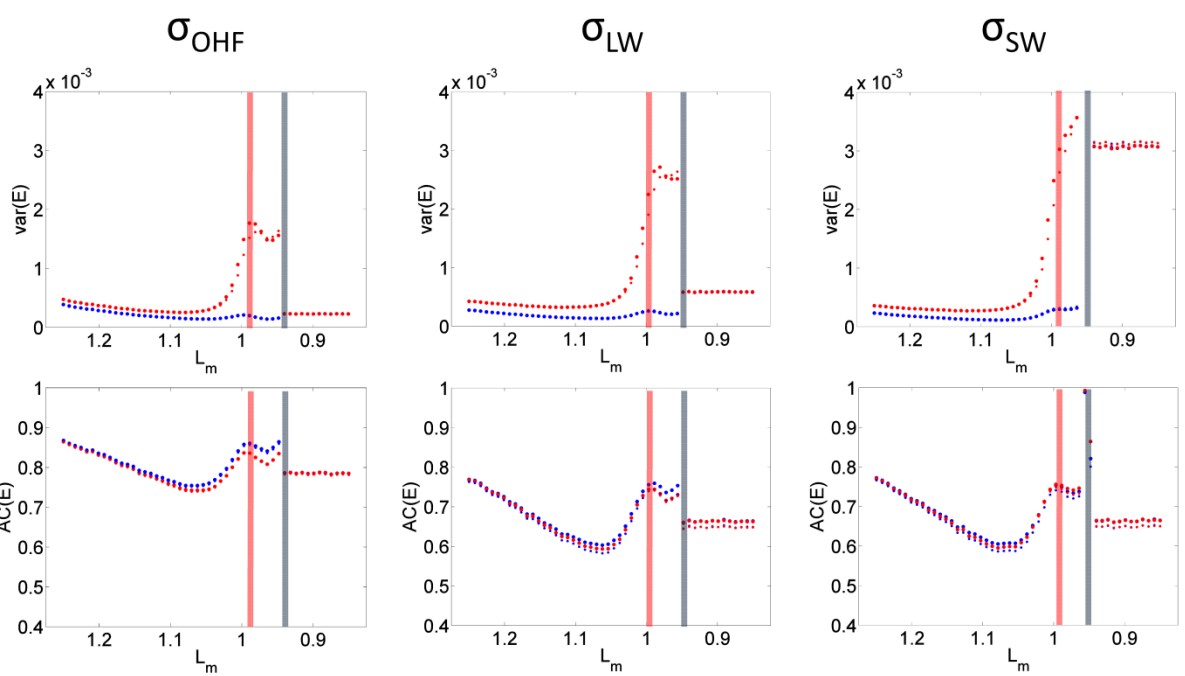

Figure 4. Variance (var) and autocorrelation (AC) of state E as a function of long-wave balance $L_m$ in the model E12 with large red noise. In each column, the noise term has been introduced to one of three different terms, namely the ocean heat flux (OHF), long-wave radiative balance (LW) and short-wave radiative balance (SW). Winter sea ice is shown in blue, summer sea ice in red (seasons as in Fig. 1). The vertical shaded lines indicate the values of $L_m$ where the annual minimum (red lines) and maximum (blue lines) ice volume reaches zero.

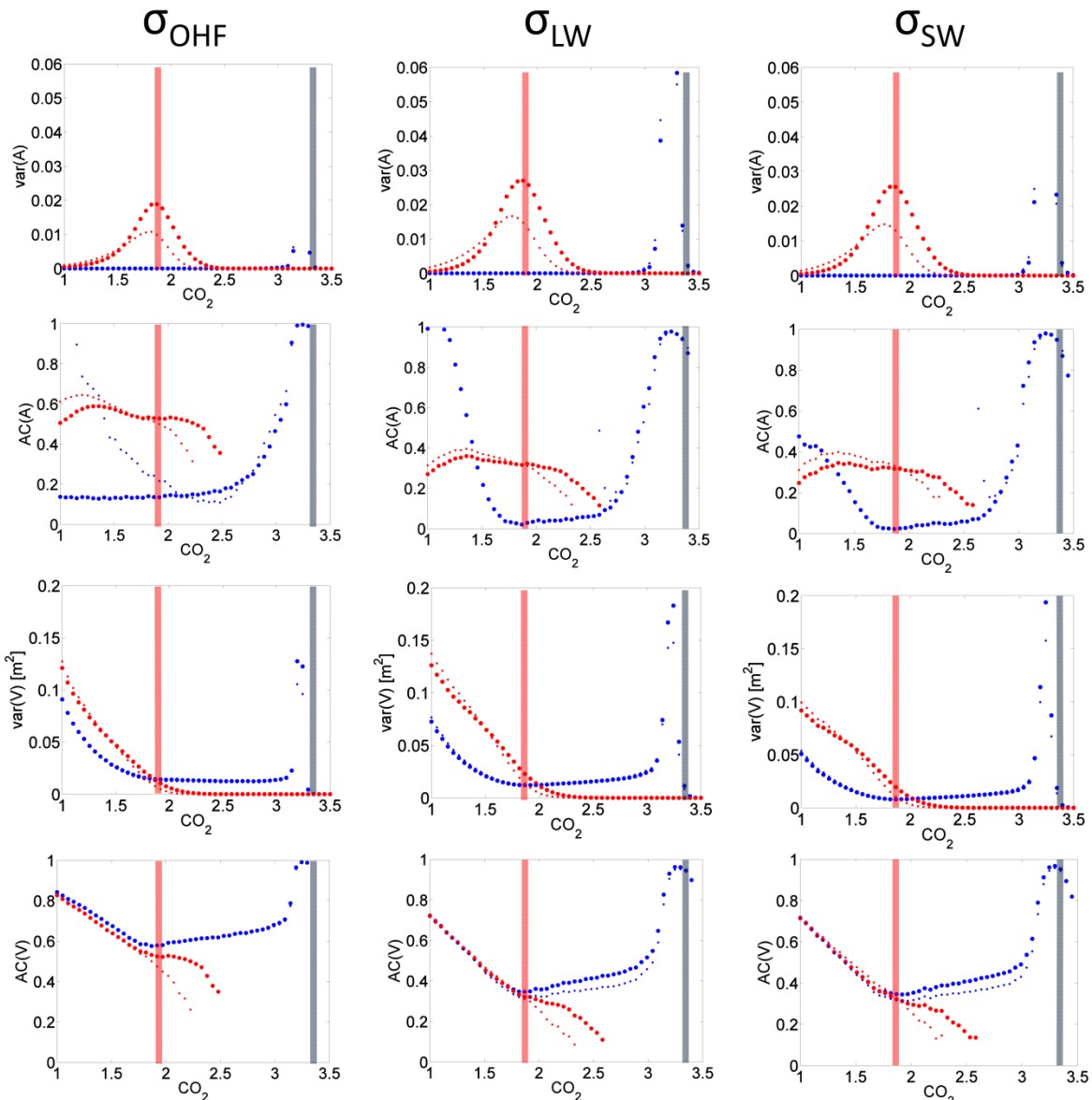

Figure 5. Variance (var) and autocorrelation (AC) of ice area fraction A and ice volume V as a function of $CO_2$ in the model E07 with large red noise. In each column, the noise term has been introduced to one of three different terms, namely the ocean heat flux (OHF), long-wave radiative balance (LW) and short-wave radiative balance (SW). Winter sea ice is shown in blue, summer sea ice in red (seasons as in Fig. 1). The vertical shaded lines indicate the values of $CO_2$ where the annual minimum (red lines) and maximum (blue lines) ice volume reaches zero.

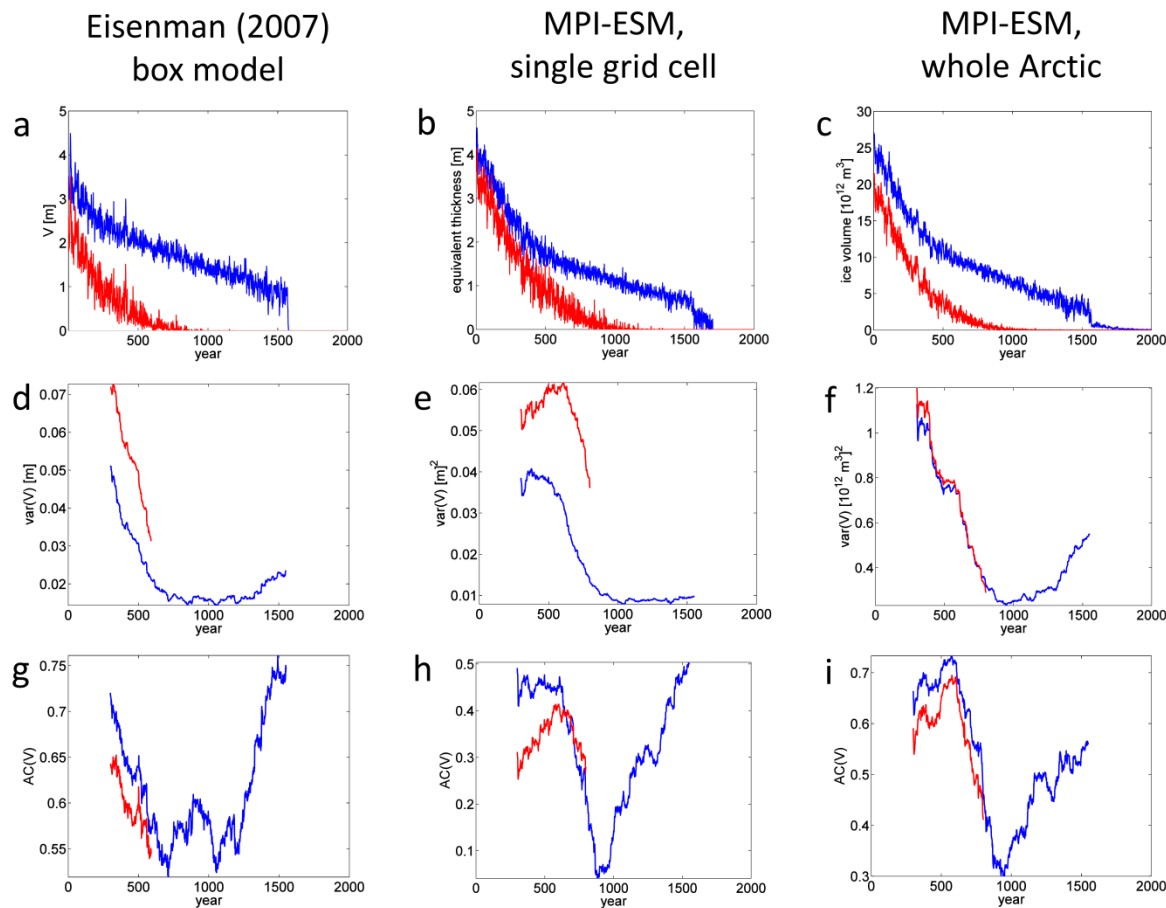

Figure 6. Time series from transient simulations with the box model by Eisenman (2007) (left column) and MPI-ESM (middle and right column). (a)-(c) evolution of ice volume; (d)-(f) variance, and (g)-(i) autocorrelation of this volume as obtained from a sliding window approach. Winter sea ice is shown in blue, summer sea ice in red. The single grid cell in MPI-ESM (second column) is located at 86.5 N and 102 W.

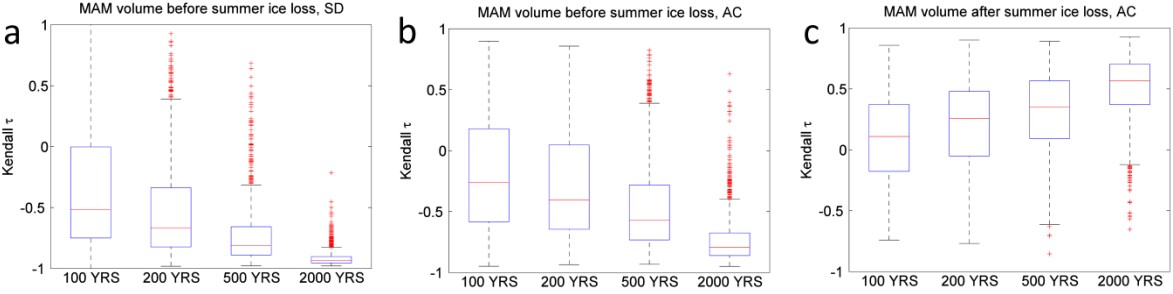

Figure 7. Statistics of Kendall's Tau for standard deviation (SD) and autocorrelation (AC) changes in ensemble simulations with the E07 model. Each figure shows results for MAM timeseries of sea-ice volume. The number of years refer to the total length of an experiment until $CO_2$ has quadrupled. a) standard deviation trends before summer sea-ice loss b) autocorrelation trends before summer sea-ice loss (perennial ice regime), c) autocorrelation trends in the period between summer sea-ice loss and winter sea-ice loss (seasonal ice regime). On each box, the central mark is the median, the edges of the box are the 25th and 75th percentiles, the whiskers extend to the most extreme data points not considered outliers, and outliers are plotted individually.

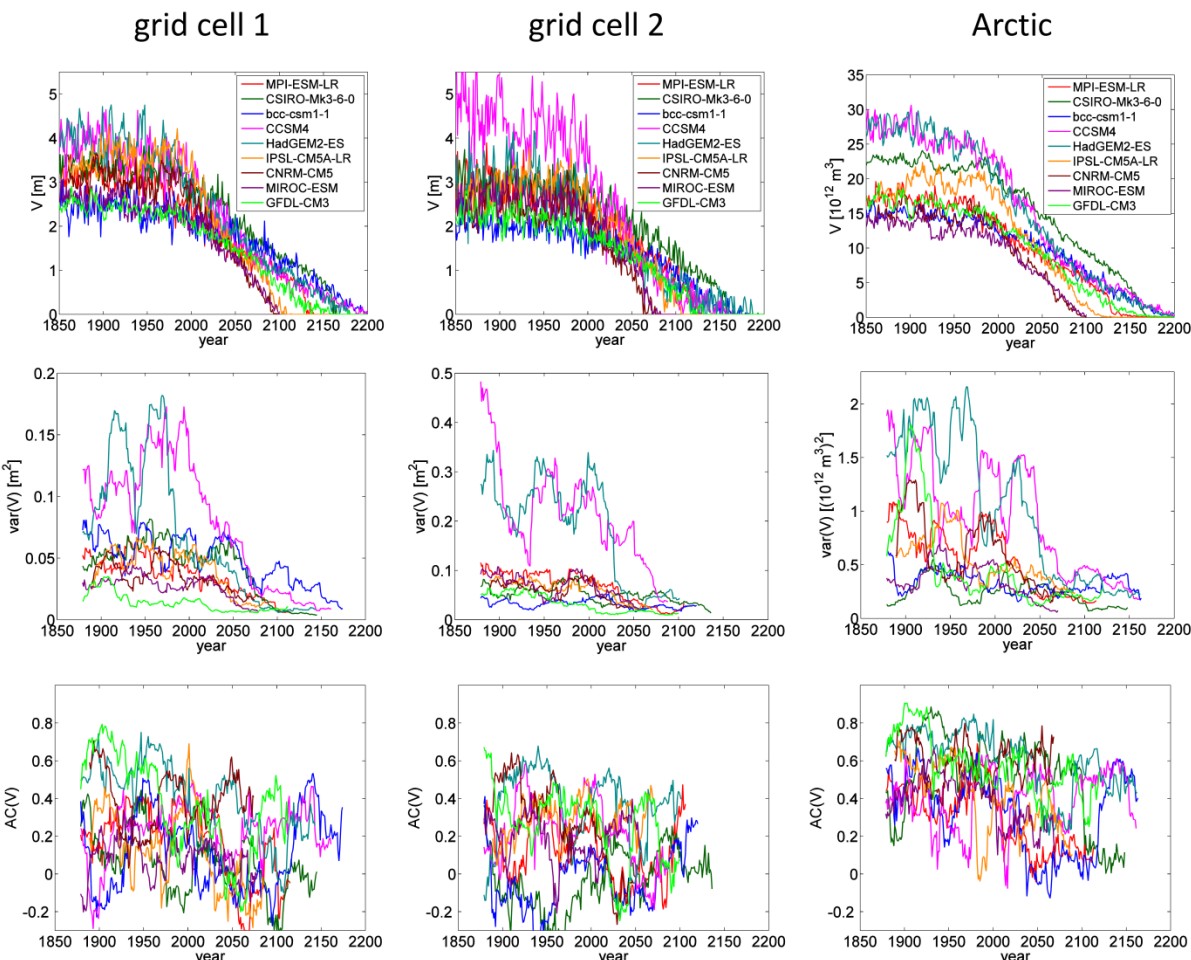

Figure 8. Evolution of MAM-averaged sea-ice volume (V) and its variance (var) and autocorrelation (AC) in nine comprehensive climate models. The time series are the combined historical and RCP8.5 simulation, the window length for the calculation of var and AC is 30 years. Grid cell 1 is located at approx. 102 W / 86.5 N, grid cell 2 at 180 W / 74.5 N, the right column shows all volume north of 75 N.