# Peer review of "Trends in sea-ice variability on the way to an ice-free Arctic"

_The Cryosphere, 2015_

## Referee Comment (RC1) · Anonymous Referee #1 · 15 Feb 2016

The paper shows a lack of familiarly with sea ice and the relevant literature. While a contribution from outside the field of sea ice modeling is generally very welcome and can lead to insights previously missed due to the application of different methods, this paper unfortunately does not provide any such insights. It stays limited to the application of an interesting method (statistical stability indicators theory), but fails to draw any useful conclusions from the analysis. The fact that a recent paper of Wegner and Eisenman (2015) explains why simple models such as EBMs and SCMs (as used in this study) tend to show instabilities and tipping points in sea ice, but complex earth system models generally do not, makes this analysis even less useful for understanding Arctic sea ice evolution in the real world (or in climate models). The paper is also build on the wrong claim that several CMIP5 models loose winter sea ice by the end of the 21st century, which is not correct. I very much regret to recommend a rejection of the

paper, as the authors clearly invested a lot of work into this contribution and wrote a well structured paper. But the factual errors, lack of awareness of the relevant literature, and the lack of any relevant conclusions does not allow me to recommend publication of this paper in The Cryosphere.

Specific comments:

1. The study by Wagner and Eisenman (2015) showed that "It is found that the stability of the ice cover vastly increases with the inclusion of spatial communication via meridional heat transport or a seasonal cycle in solar forcing, being most stable when both are included." And "the present model simulates sea ice loss that is not only reversible but also has a strikingly linear relationship with the climate forcing as well as with the global-mean temperature. This is in contrast with SCMs and EBMs, and it is consistent with GCMs. The results presented here indicate that the nonlinearities in the model are essentially smoothed out when latitudinal and seasonal variations are included. " This important study was not cited, despite the fact that it was published over a year ago (in Feb 2015). As the authors are using SCMs and EBMs to study instabilities in the sea ice system, and Wagner and Eisenman showed that these models overestimate instabilities due to their lack of a spatial dimension, this paper removes the basis of the work presented here.

2. Even without this recent paper, the title is misleading, as no physical insights into "Trends in sea-ice variability on the way to an ice-free Arctic" are shown. If anything, the title should be "Relaxation time and autocorrelation in the Arctic sea ice cover on the way to an ice-free Arctic" or "Statistical stability indicators theory applied to Arctic sea ice", as the study focused only on the application of the method, without providing physical insights into the actual system of sea ice decline.

3. Page 4, line 18-22: "The models are all the available models that lose their Arctic winter sea ice in RCP8.5". This is wrong. I don't know of any CMIP5 models that lose their Arctic winter sea ice in RCP8.5 by 2100. Some of them do by the end of the 24th

century in the extended concentration pathway scenarios (see Hezel et al. 2014), but not by the end of the 21st century. So if the authors wanted to study the winter sea ice going away in GCM simulations, the extended concentration pathway simulations (shown in Hezel et al. 2014) would need to be used. Also, why is the sea ice volume time series not shown in Figure 8? It would show that these models do not lose winter sea ice under RCP8.5 by the end of the 21st century simulations, so the authors should have plotted it to avoid this mistake.

4. Page 9, line 35: "the inclusion of spatial differences and processes like advection and mechanical redistribution of sea ice apparently has not changed the behavior of sea ice variability. We therefore argue that E07 is an appropriate model to explain the behavior in MPI-ESM and it is probable that the same processes are behind the evolution of the statistics."

5. This statement is in direct conflict with Wagner and Eisenman (2015), and therefore needs further investigation. Maybe the MPI model is an outlier in the CGM's that participated in the CMIP5, due to its very simple sea ice model (compared to the other GCMs in CMIP5)? The authors would need to present results from more than one CGM in order to be able to make the results robust. There are many other models that have run 4xCO2 experiments for CMIP5; the authors would need to analyze these to show that the MPI model is not an outlier in that it shows a abrupt transition in winter sea ice, as Henzel et al 2014 does not show any CMIP5 models showing abrupt transitions to winter ice free in the extended RCP8.5 simulations. Furthermore, "abrupt" is not defined anywhere in the paper. The "rapid" ice loss shown in the MPI model occurs at a relatively small Arctic ice volume, so it does not constitute a large change or big transition in any case.

References: Till J. W. Wagner and Ian Eisenman, 2015: How Climate Model Complexity Influences Sea Ice Stability. J. Climate, 28, 3998–4014. doi: http://dx.doi.org/10.1175/JCLI-D-14-00654.1

[Figure]

Hezel, P. J., Fichefet, T., and Massonnet, F.: Modeled Arctic sea ice evolution through 2300 in CMIP5 extended RCPs, The Cryosphere, 8, 1195-1204, doi:10.5194/tc-8-1195-2014, 2014.
* * *

---

## Author Comment (AC1) · 17 Feb 2016

We are grateful to the reviewer for his constructive criticism, and in particular for pointing out two references that we should have cited in our publication. We will do so in a revised version of this manuscript, and we will clarify the relationship between our work and these two existing studies:

1. We do not assume the existence of a bifurcation in Arctic sea ice. A main result of our paper is that we demonstrate and explain a robust link between the mean state and the variability of sea ice. As we show, this link does not depend on whether there is a bifurcation or not, neither does it matter how abruptly the ice loss occurs. For example, the abrupt winter sea-ice loss in the model by Eisenman (2007) is not due to a bifurcation, and it is not in conflict with the findings of Wagner and Eisenman (2015).

[Figure]

The mechanism in Eisenman (2007) resembles the one found in complex models. We demonstrate this in a recent paper where we also show that Arctic winter sea-ice loss is more rapid than the preceding summer sea-ice loss in all models where it occurs (Bathiany et al., 2016). MPI-ESM is no outlier in terms of the underlying mechanism, and we will clarify this point in the revised version.

2. We do use the extended RCP8.5 simulations, and this can be seen in Fig. 8 where most time series extend beyond 2100. As Hezel et al. (2014) point out in their abstract, "In RCP8.5 the Arctic Ocean reaches annually ice-free conditions in seven of nine models." We simply adopted their notation of calling the extended RCP8.5 by their initial name. We do not claim that the models would lose their winter ice before 2100. This is only true for two of the models (GFDL-CM3 and MIROC-ESM) where the Arctic winter ice area drops below one million square kilometers. Moreover, our results do not depend on the timing of the winter ice loss. In a revised version, in addition to clarifying these points, we will include a plot of sea-ice volume time series, as suggested by the reviewer.

References

Bathiany, S., D. Notz, T. Mauritsen, G. Raedel, and V. Brovkin, 2016: On the potential for abrupt Arctic winter sea-ice loss. J. Climate. doi:10.1175/JCLI-D-15-0466.1, in press.

Eisenman, I.: Arctic catastrophes in an idealized sea-ice model, in: 2006 Program of studies: Ice (geophysical fluid dynamics program), 133-161, Woods Hole Oceanographic Institution, Woods Hole, Mass., 2007.

Hezel, P. J., T. Fichefet, and F. Massonnet, 2014: Modeled Arctic sea ice evolution through 2300 in CMIP5 extended RCPs. Cryosphere, 8, 1195-1204, doi:10.5194/tc-8-1195-2014.

Wagner, T., and I. Eisenman, 2015: How Climate Model Complexity Influences Sea Ice

Stability. J. Climate, 28, 3998-4014, doi:10.1175/JCLI-D-14-00654.1.

---

## Referee Comment (RC2) · Anonymous Referee #1 · 18 Feb 2016

As the answer to my review raises points that are still problematic in my opinion, a few new comments:

In reply to point 1, Hezel et al. (2014) find that "in all but two models, however, sea ice volume demonstrates a continuing linear or slower rather than faster rate of decline through the disappearance of winter ice, and thus we conclude that apparent threashold behavior is not occurring in this set of models as the winter sea ice disappears". With the MPI-ESM-LR model being the model that shows the most notable non-linear decline in sea ice towards an ice-free state. Which is opposite to the claim of the authors that "MPI-ESM is no outlier in terms of the underlying mechanism, and we will clarify this point in the revised version", and supports my concern that the MPI model is not the right model to use for this study, as it behances differently than other CMIP5 GCMs.
[Figure]

In regards to point 2, the authors description of the CMIP5 model simulations they used did not at all reflect that they used the extended concentration pathway simulations (but I can see now that the lines in Fig 8 extend past 2100). The use of the term RCP8.5 (which describes simulations from 2005-2100), and the reference that "reaching a radiative forcing of approximately 8.5 Wm2 in the year 2100" directly before the statement that these models all loose their winter sea ice in RCP8.5 is very misleading, and also shows a lack of familiarity with the CMIP5 models/scenarios (also shown in the absence of any references for these scenarios/simulations, which could have clarified the text for the informed reader). Hezel et al. (2014) used RCP8.5 to refer to the continous simulations (2005-2300), but clearly explained what they were doing and cited the relevant literature, which was both not done here and needs to be improved upon greatly if the editor decides to request a revised submission. The relevant papers for the extended concentration pathway experiments the authors used are Moss et al. (2010) and Meinshausen et al. (2011).
* * *

---

## Referee Comment (RC3) · Anonymous Referee #2 · 19 Feb 2016

This paper investigates temporal characteristics of Arctic sea ice area and volume under increased radiative forcing, in two box models and a comprehensive coupled atmosphere ocean sea ice model. Previous literature has investigated the possibility that, due to the positive ice-albedo feedback, sea ice could reach a tipping (or bifurcation point), and whether or not this would be preceded by increased auto-correlation and variance as predicted by theoretical models. Prior to summer sea ice loss, the response time found to decrease, as thinner ice can adjust more quickly to perturbations. Between summer and annual sea ice loss the response time is found to increase, as the system is more constrained by the large heat capacity of the ocean. Switching off individual mechanisms leads in some cases to a loss of a bifurcation point, but otherwise similar characteristics in response time.

In general, I find the analysis presented convincing and technically sound, but I share

the concerns that have also been expressed by the other reviewer. Specifically:

1) It is hard to judge the relevancy of this work for the actual world. Wagner and Eisenman shows that if you include meridional heat transport (a mechanism not included in the box models considered in the current study) the non-linearity from albedo changes is effectively removed and no tipping point is found to occur. Hence, the box models considered here are likely too simple to be relevant to the real world.

2) It's also hard to judge the novelty of results presented in the current study. Which aspects of the results are novel, and which are simply confirmations of results already published in previous literature (such as the two Wagner and Eisenman papers)? On page 3, line 14 the authors state that 'it has not been investigate how these factors affect the prospects for early warning signals, especially in more complex, spatially explicit models...'. In the previous sentence, the authors state that Wagner and Eisenman have investigated this issue...

3) P. 6, line 29. Even though Wagner and Eisenman also find the lack of a bifurcation point in their model, this seems to be the case for a fundamentally different reason. In their case, they increased the complexity of their model (by including meridional heat transport), whereas here you decreased it.

4) The implications for other systems are unclear to me. The presented results seem to be very specific to sea ice area and the specific feedback processes relevant for sea ice.

5) P.9: For easier interpretation it would be helpful if you could quote the $CO_2$ quadrupling time time in extended RCP 8.5 simulations.

———————————————

---

## Short Comment (SC1) · 7 Mar 2016

In agreement with, and in addition to, the insightful and constructive comments of the reviewers, I would like to provide the following feedback to this interesting study, which I think will make a valuable contribution to the literature:

1) Overall, this paper provides interesting and novel insight into the statistical differences between summer and winter sea ice loss, as well as the evolution of sea ice thickness and volume. It therefore goes beyond previous work, including our paper (Wagner & Eisenman, 2015), where we focused on sea ice area during summer. It further fills in important gaps regarding the effects of different types of stochastic forcing.

2) Title and introduction: Regarding the title, I agree broadly with Reviewer 2 that it may be better if the title referred specifically to the statistical indicators, since they are

at the core of this study. Regarding the introduction, I would proffer that the focus could be shifted somewhat toward the evolution of variance and autocorrelation under sea ice loss in general, rather than focusing on their (lack of) utility as early warning signals for critical transitions. Introducing the concept of using variance and autocorrelation to help estimate the future mean state and variability of the sea ice cover is an excellent contribution of this study that could be given more weight here in my opinion.

3) Relatedly, a slightly clearer presentation of what has been published on this topic and what is novel, in line with comment 2 by Reviewer 2, may improve the exposition of the paper.

4) As a side note, I want to point out that the spatially explicit model results from Wagner & Eisenman (2015) show an increase in autocorrelation before the loss of the summer sea ice, something that single-column models like E07 may not always pick up on. We suggest that the increase in autocorrelation is due to the growth of the (long-memory) open-water region as the ice retreats, in agreement with the conclusions drawn here.

5) The analysis and discussion of GCM results appears to me (not a GCM expert) very valuable. It highlights a number of important operational limitations in applying statistical indicators as early warning signals, and it provides the first steps toward the use of statistical indicators in GCMs to predict changes in the sea ice cover. I would hope this motivates fruitful further research in the community.

Reference: T.J.W. Wagner and I. Eisenman (2015) "False alarms: How early warning signals falsely predict abrupt sea ice loss", GRL (23) 42, DOI: 10.1002/2015GL066297

---

## Author Response (AR1)

Reviewer #1, comment 1:

The paper shows a lack of familiarly with sea ice and the relevant literature. While a contribution from outside the field of sea ice modeling is generally very welcome and can lead to insights previously missed due to the application of different methods, this paper unfortunately does not provide any such insights. It stays limited to the application of an interesting method (statistical stability indicators theory), but fails to draw any useful conclusions from the analysis. The fact that a recent paper of Wegner and Eisenman (2015) explains why simple models such as EBMs and SCMs (as used in this study) tend to show instabilities and tipping points in sea ice, but complex earth system models generally do not, makes this analysis even less useful for understanding Arctic sea ice evolution in the real world (or in climate models). The paper is also build on the wrong claim that several CMIP5 models loose winter sea ice by the end of the 21st century, which is not correct. I very much regret to recommend a rejection of the paper, as the authors clearly invested a lot of work into this contribution and wrote a well structured paper. But the factual errors, lack of awareness of the relevant literature, and the lack of any relevant conclusions does not allow me to recommend publication of this paper in The Cryosphere.

Specific comments:
1. The study by Wagner and Eisenman (2015) showed that "It is found that the stability of the ice cover vastly increases with the inclusion of spatial communication via meridional heat transport or a seasonal cycle in solar forcing, being most stable when both are included." And "the present model simulates sea ice loss that is not only reversible but also has a strikingly linear relationship with the climate forcing as well as with the global-mean temperature. This is in contrast with SCMs and EBMs, and it is consistent with GCMs. The results presented here indicate that the nonlinearities in the model are essentially smoothed out when latitudinal and seasonal variations are included. " This important study was not cited, despite the fact that it was published over a year ago (in Feb 2015). As the authors are using SCMs and EBMs to study instabilities in the sea ice system, and Wagner and Eisenman showed that these models overestimate instabilities due to their lack of a spatial dimension, this paper removes the basis of the work presented here.

2. Even without this recent paper, the title is misleading, as no physical insights into "Trends in sea-ice variability on the way to an ice-free Arctic" are shown. If anything, the title should be "Relaxation time and autocorrelation in the Arctic sea ice cover on the way to an ice-free Arctic" or "Statistical stability indicators theory applied to Arctic sea ice", as the study focused only on the application of the method, without providing physical insights into the actual system of sea ice decline.

3. Page 4, line 18-22: "The models are all the available models that lose their Arctic winter sea ice in RCP8.5". This is wrong. I don't know of any CMIP5 models that lose their Arctic winter sea ice in RCP8.5 by 2100. Some of them do by the end of the 24th century in the extended concentration pathway scenarios (see Hezel et al. 2014), but not by the end of the 21st century. So if the authors wanted to study the winter sea ice going away in GCM simulations, the extended concentration pathway simulations (shown in Hezel et al. 2014) would need to be used. Also, why is the sea ice volume time series not shown in Figure 8? It would show that these models do not lose winter sea ice under RCP8.5 by the end of the 21st century simulations, so the authors should have plotted it to avoid this mistake.

4. Page 9, line 35: "the inclusion of spatial differences and processes like advection and mechanical redistribution of sea ice apparently has not changed the behavior of sea ice variability. We therefore argue that E07 is an appropriate model to explain the behavior in MPI-ESM and it is probable that the same processes are behind the evolution of the statistics."

5. This statement is in direct conflict with Wagner and Eisenman (2015), and therefore needs further investigation. Maybe the MPI model is an outlier in the CGM's that participated in the CMIP5, due to its very simple sea ice model (compared to the other GCMs in CMIP5)? The authors would need to present results from more than one

CGM in order to be able to make the results robust. There are many other models that have run 4xCO2 experiments for CMIP5; the authors would need to analyze these to show that the MPI model is not an outlier in that it shows a abrupt transition in winter sea ice, as Henzel et al 2014 does not show any CMIP5 models showing abrupt transitions to winter ice free in the extended RCP8.5 simulations. Furthermore, "abrupt" is not defined anywhere in the paper. The "rapid" ice loss shown in the MPI model occurs at a relatively small Arctic ice volume, so it does not constitute a large change or big transition in any case.

We thank the referee for the constructive comments.
The first comment raises the concern that the study by Wagner and Eisenman (2015a) could affect the relevance of our study. Wagner and Eisenman show that tipping points can occur as a model artefact in simple models (EBMs and SCMs) because the seasonal cycle and spatial differences are not resolved properly. We now cite this important paper in the introduction of our revised article. However, our article does not make any assumptions on the existence of a tipping point, but focuses on the relation between the mean state and the variability of sea ice, before sea ice is lost completely. By doing so, we assess if statistical stability indicators can predict a potential tipping point. Such an analysis is useful because observations might then provide an additional source of information about sea ice stability, besides the predictions of climate models that are always uncertain so some extent. Moreover, multiple steady states have also been found in complex models. The latter results are not directly relevant for the loss of sea ice in the coming centuries, but they are potentially important to understand past climate change. We understand that we should have made these arguments more specific and have revised the manuscript accordingly. In particular, we point out in the introduction:

*"Wagner and Eisenman (2015a) recently showed in detail how resolving the seasonal cycle and latitudinal differences can eliminate bifurcations in sea-ice models. Nonetheless, bifurcations also occur in comprehensive climate models: In a complex general-circulation model with current continental distribution and solar insolation, Marotzke and Botzet (2007) identified a globally ice-covered stable state analogous of the 'Snowball Earth' conditions in the Neoproterozoic (Pierrehumbert et al., 2011). Ferreira et al. (2011) and Rose et al. (2013) even found three stable states in a complex model with idealised ocean geometry. Such alternative stable states imply the possibility of large-scale abrupt climate changes when external conditions are varied. Moreover, Ferreira et al. (2011) and Rose et al. (2013) show that the existence of multiple stable sea-ice states depends on the structure of the ocean circulation, a nonlinear system that can even show tipping point behaviour on its own. Such nonlinear interactions are not captured by the model of Wagner and Eisenman (2015a) because heat transport is formulated as a simple diffusion term in their model which has only one spatial dimension. Given these model uncertainties, it is worthwhile to investigate the changes in variability that are associated with sea-ice loss, mainly for two practical reasons. First, if these changes depend on the abruptness of future sea-ice loss, observations might provide an alternative source of information and indicate which model is most reliable in its prediction. Second, one might draw conclusions about the climate variability and the rates of change in the Earth's deep past, something that is difficult to reconstruct directly (White et al., 2010; Kemp et al., 2015), and that can help to build simple stochastic climate models.."*

As we already pointed out in our previous reply, we do not make any (false) claim about when Arctic winter sea ice would be lost in the models.

Specific comments:

1. As noted above, we now cite the paper by Wagner and Eisenman (2015a), and we explain why our study is not in conflict with their results.

2. We do provide physical insights, in particular in Sect. 3.1 where we demonstrate the physical reason for the decrease in time scale during summer ice melt (growth-thickness feedback), and the increase in time scale during winter ice melt (mixed-layer effect). Although the existence of these effects is already known, it has not been tested before if they would also dominate sea-ice

variability in comprehensive models. Our study investigates this question for the first time. As the link between mean state and variability proves robust in the models, we think that the title is not misleading. It is true that we focus on ice volume in the paper because several papers have been published about the variance of ice area, and because the autocorrelation of ice area shows no clear trends (as we mention in the paper). We have decided to not make the title too technical and mention these details in the abstract and the rest of the paper.

3. We now show the time series of sea-ice volume in Fig. 8. These figures and a revised methods section make clearer that we do indeed also analyse the extended RCP8.5 scenario. We now also refer to Hezel et al. (2014) in this section.

4./5. Our revised manuscript points out more clearly that we do analyse several Earth system models, though MPI-ESM is indeed analysed in most detail. The fact that the model by Eisenman (2007) can explain the behaviour of MPI-ESM is confirmed by a previous study (Bathiany et al., 2016) which we now cite. The reviewer has also raised concerns about the realism of our results given the abrupt ice loss in MPI-ESM compared to other CMIP5 models. In our revised manuscript we explain more clearly that our analysis concerns the changes in sea-ice variability that occur before the final loss of winter sea ice, and that these changes do not depend on how abrupt this final ice loss is. In particular, we have added a paragraph in Sect. 4 (Conclusions) to explain why the MPI-ESM is not an outlier in terms of its representation of sea-ice variability:

*"The comprehensive model we analysed in most detail, MPI-ESM, likely exaggerates how rapidly the final bit of winter sea-ice volume disappears (e.g. as seen in the top right panel of Fig. 8). This abrupt volume loss is probably related to the ice-growth parameterisation, which attributes a single thickness to all newly formed ice in a grid cell (Bathiany et al., 2016). Although the abrupt event itself is not part of our time series analysis above, it points to potential limitations of the applied model and one may ask how models with several ice-thickness classes would behave. It is reassuring in this regard that eight other models agree with MPI-ESM in their decrease of the sea-ice volume's variance, although time series were too short to show clear trends in autocorrelation. Moreover, the mechanistic insight obtained with the simpler models suggests that these model agreements are no coincidence because they can be explained from fundamental physical processes. Both the fast adjustment of thin ice and the slow response of the mixed-layer ocean are represented in all the models and would also not change in even more complex models. For example, in models with many ice-thickness classes, the variability of the total ice volume in a grid cell is the result of the variability of all thickness classes. The trends in variance and autocorrelation would have the same sign for each thickness class because the thickness-growth relationship is monotonous (Thorndike et al., 1975). Even the precise realisation of the weather-induced variability would be identical because all thickness classes within a grid cell are coupled to the same ocean and atmosphere grid cell. Hence, the level of sophistication in the representation of the subgrid-scale ice-thickness distribution is not relevant for our results. Furthermore, it has been shown in Bathiany et al. (2016) that radiative feedbacks and mechanical redistribution mechanisms are unimportant for the abruptness of sea-ice loss in MPI-ESM, which is instead determined by thermodynamic processes. It is therefore plausible that the same processes also determine the variability of sea ice before the final ice loss occurs."*

Following the reviewer's suggestion, we now also define abrupt change in the introduction:
*"Such a change is loosely referred to as 'abrupt' if the acceleration is due to mechanisms internal to the climate system (such as the positive ice-albedo feedback) whereas the forcing changes linearly over time (Rahmstorf, 2001; National Research Council, 2002)."*
We do not use the word rapid anymore in this context.

Bathiany, S., Notz, D., Mauritsen, T., Brovkin, V., and Raedel, G.: On the potential for abrupt Arctic winter sea-ice loss, J. Clim., 2016.

Eisenman, I.: Arctic catastrophes in an idealized sea-ice model, in: 2006 Program of studies: Ice (geophysical fluid dynamics program), 133-161, Woods Hole Oceanographic Institution, Woods Hole, Mass., 2007.

Ferreira, D., Marshall, J., and Rose, B.: Climate determinism revisited: multiple equilibria in a complex climate model. J. Climate, 24, 992-1012, 2011.

Hezel, P. J., Fichefet, T., and Massonnet, F., 2014: Modeled Arctic sea ice evolution through 2300 in CMIP5 extended RCPs. Cryosphere, 8, 1195-1204, 2014.

Kemp, D. B., Eichenseer, K., and Kiessling, W.: Maximum rates of climate change are systematically underestimated in the geological record. Nat. Commun., 6, 8890, 2015.

Marotzke, J., and Botzet, M.: Present-day and ice-covered equilibrium states in a comprehensive climate model. Geophys. Res. Lett., 34, L16704, 2007.

National Research Council: Abrupt Climate Change: Inevitable Surprises, Natl. Acad. Press, Washington, DC, 2002.

Pierrehumbert, R. T., Abbot, D. S., Voigt, A., and Koll, D.: Climate of the Neoproterozoic, Annu. Rev. Earth Planet. Sci., 39, 417–460, 2011.

Rahmstorf, S.: Abrupt Climate Change. Encyclopedia of Ocean Sciences, eds Steele J, Thorpe S, Turekian K (Academic, London), pp 1–6, 2001.

Rose, B., Ferreira, D., and Marshall, J.: The role of oceans and sea ice in abrupt transitions between multiple climate states. J. Climate, 26, 2862-2879, 2013.

Thorndike, A. S., Rothrock, D. A., Maykut, G. A., and Colony, R.: The thickness distribution of sea ice. J. Geophys. Res., 80, 4501-4513, 1975.

Wagner, T. J. W., and Eisenman, I.: How climate model complexity influences sea ice stability. J. Climate, 28, 3998-4014, 2015a.

White, J. W. C., and Coathors: Past rates of climate change in the Arctic. Quat. Sci. Rev., 29, 1716-1727, 2010.

Reviewer #1, comment 2:

"In reply to point 1, Hezel et al. (2014) find that "in all but two models, however, sea ice volume demonstrates a continuing linear or slower rather than faster rate of decline through the disappearance of winter ice, and thus we conclude that apparent threshold behavior is not occurring in this set of models as the winter sea ice disappears".
With the MPI-ESM-LR model being the model that shows the most notable non-linear decline in sea ice towards an ice-free state. Which is opposite to the claim of the authors that "MPI-ESM is no outlier in terms of the underlying mechanism, and we will clarify this point in the revised version", and supports my concern that the MPI model is not the right model to use for this study, as it behances differently than other CMIP5 GCMs.

In regards to point 2, the authors description of the CMIP5 model simulations they used did not at all reflect that they used the extended concentration pathway simulations (but I can see now that the lines in Fig 8 extend past 2100). The use of the term RCP8.5 (which describes simulations from 2005-2100), and the reference that "reaching a radiative forcing of approximately 8.5 Wm2 in the year 2100" directly before the statement that these models all loose their winter sea ice in RCP8.5 is very misleading, and also shows a lack of familiarity with the CMIP5 models/scenarios (also shown in the absence of any references for these scenarios/simulations, which could have clarified the text for the informed reader). Hezel et al. (2014) used RCP8.5 to refer to the continous simulations (2005-2300), but clearly explained what they were doing and cited the relevant literature, which was both not done here and needs to be improved upon greatly if the editor decides to request a revised submission. The relevant papers for the extended concentration pathway experiments the authors used are Moss et al. (2010) and Meinshausen et al. (2011)."

We thank the referee again for these constructive comments.
Regarding the concern about the realism of MPI-ESM, see our reply above.

Concerning the simulations we analyse, we now explain them and the selection of these simulations more explicitly. In addition, we have obtained model output from one additional comprehensive model (bcc-csm1-1) that we now also analyse.

In the introduction we now write:

*"We also analyse eight additional comprehensive models from the Coupled Model Intercomparison Project 5 (CMIP5), using simulations of the historical period, the RCP8.5 scenario and its extension until the year 2300. The models are all the available models that lose their Arctic winter sea ice in these simulations. The level of complexity in these models is comparable to MPI-ESM, but some of them explicitly resolve several ice-thickness classes on the subgrid scale. Although one of the models (CSIRO-Mk3-6-0) also produces an abrupt loss of winter sea-ice area, most models show a retreat of winter sea ice that is gradual (Hezel et al., 2014), though faster than the preceding summer sea-ice loss (Bathiany et al., 2016)."*

In addition, we elaborate on this in Sect. 3.3:

*"To test this prediction, we finally analyse CMIP5 simulations from MPI-ESM and eight other comprehensive climate models. For this analysis we combine the historical simulation, the RCP8.5 simulation, and the extended RCP8.5 simulation that ends in the year 2300. In this scenario, atmospheric $CO_2$ shows an accelerated increase until the year 2100, when a radiative forcing of approx. 8.5 W/m2 is reached. Thereafter, the $CO_2$ concentration stabilises at almost 2000 ppm (Meinshausen et al., 2011), yielding the largest warming of all CMIP5 simulations. The extended simulations until 2300 were performed with nine models (Hezel et al., 2014). Here we analyse all models where Arctic sea-ice area falls below one million square kilometres in the full RCP8.5 scenario, no matter when this event occurs. Two of the models analysed in Hezel et al. (2014) do not lose their winter sea ice by 2300, while two other models not analysed by Hezel et al. (2014) have lost their winter sea ice already by 2100 (the nine models we analyse are therefore not identical to the nine models in Hezel et al., 2014)."*

While this includes a reference to Meinshausen et al (2011) as suggested by the reviewer, we do not cite the paper by Moss et al. (2010) because it does not discuss the extended RCP8.5 simulation.

Bathiany, S., Notz, D., Mauritsen, T., Brovkin, V., and Raedel, G.: On the potential for abrupt Arctic winter sea-ice loss, J. Clim., 2016.

Hezel, P. J., Fichefet, T., and Massonnet, F., 2014: Modeled Arctic sea ice evolution through 2300 in CMIP5 extended RCPs. Cryosphere, 8, 1195-1204, 2014.

Meinshausen, M., and Coauthors: The RCP greenhouse gas concentrations and their extensions from 1765 to 2300. Climatic Change, 109, 213–241, 2011.

Moss, R. H., and Coauthors: The next generation of scenarios for climate change research and assessment. Nature, 463, 747-756, 2010.

Reviewer #2:

In general, I find the analysis presented convincing and technically sound, but I share the concerns that have also been expressed by the other reviewer. Specifically:

1) It is hard to judge the relevancy of this work for the actual world. Wagner and Eisenman

shows that if you include meridional heat transport (a mechanism not included in the box models considered in the current study) the non-linearity from albedo changes is effectively removed and no tipping point is found to occur. Hence, the box models considered here are likely too simple to be relevant to the real world.

2) It's also hard to judge the novelty of results presented in the current study. Which aspects of the results are novel, and which are simply confirmations of results already published in previous literature (such as the two Wagner and Eisenman papers)? On page 3, line 14 the authors state that 'it has not been investigate how these factors affect the prospects for early warning signals, especially in more complex, spatially explicit models...'. In the previous sentence, the authors state that Wagner and Eisenman have investigated this issue...

3) P. 6, line 29. Even though Wagner and Eisenman also find the lack of a bifurcation point in their model, this seems to be the case for a fundamentally different reason. In their case, they increased the complexity of their model (by including meridional heat transport), whereas here you decreased it.

4) The implications for other systems are unclear to me. The presented results seem to be very specific to sea ice area and the specific feedback processes relevant for sea ice.

5) P.9: For easier interpretation it would be helpful if you could quote the CO2 quadrupling time time in extended RCP 8.5 simulations.

We thank the referee for these constructive comments which helped us to improve the manuscript.

1. We see two major aspects in our study that are relevant for reality. First, the robust link between mean state and variability of sea ice is useful to know in order to infer the variability of sea ice in future and past climates. For example, our results would allow to formulate a simple stochastic parameterisation of sea-ice variability. Second, we assess the performance of statistical stability indicators that are sometimes applied to observations and reconstructions. It is often argued that the method could provide information on climate stability, independently of any complex model. However, the success of the theory is usually only demonstrated in very simple stochastic models. In more complex systems, there can be many counteracting effects, and it is not self-evident if a simple one-dimensional theory holds in a complex world. Therefore, it is necessary to investigate if the approach can yield meaningful results in the case of Arctic sea ice, and how the results depend on the model formulation and complexity.

We agree with recent studies that Arctic sea ice is probably not approaching a tipping point. However, given the model uncertainties such projections are never completely certain. Our study shows that if sea ice was approaching a tipping point, observations of sea-ice variability would not help to detect it. Hence, we indeed do have to trust the models, but we think that it is useful to know this.

We have revised the introduction and conclusions sections of our study to point out these aspects more clearly.

2. Our study is novel in mainly two aspects. First, it is more comprehensive than previous studies by analysing and interpreting variability between the states of perennial ice cover and an ice-free ocean. In contrast, Moon and Wettlaufer (2011, 2013) did not analyse variability at all, whereas Wagner and Eisenman (2015b) only focussed on the mixed-layer effect. We show that statistical stability indicators do not work either in other regimes.

Second, previous studies only used simple models, the most complex being the model by Wagner and Eisenman (2015a). This model is based on the single column model by Eisenman and Wettlaufer (2009) which only predicts one state variable (enthalpy). The additional complexity Wagner and Eisenman (2015b) included in the model was to couple many 'single columns' together with a simple heat diffusion term and in only one spatial dimension (latitude). Their model describes an aquaplanet without any continents, and does not resolve an open-water fraction at the subgrid scale, which can have consequences for the heat flux between ocean and atmosphere and thus the adjustment to perturbations. Their model is therefore still much simpler than the general-circulation models used in CMIP5.

In a nutshell, we go beyond previous studies by explicitly demonstrating how sea-ice variability can be explained in the complete range of climate regimes. And, for the first time, we also analyse statistical stability indicators of sea ice in comprehensive climate models. Again, we refer the referee to our revised introduction and conclusions where we point out these aspects more clearly.

3. The part of text the reviewer refers to explains why the relaxation time of sea ice increases while seasonal sea-ice is lost. Our paper in general, and the mentioned paragraph in particular, do not analyse under what conditions bifurcations occur or do not occur. What we show here is that the system approaches the mixed-layer ocean's time scale when $CO_2$ is increased. We do this in the mentioned paragraph by directly changing this time scale in the model. This has nothing to do with the existence of the bifurcation that occurs at the transition to an ice-free ocean, and changing the mixed-layer time scale does not affect this bifurcation. The result is also not in conflict with the model of Wagner and Eisenman (2015a,b), which is another version of the model we discuss in the text (only that it has a spatial dimension), and which shows the same phenomenon. It is a main point of our paper that all models agree on this phenomenon despite their disagreement on the abruptness of the last bit of sea ice.

4. Of course, the physical mechanisms we analyse in the paper are restricted to sea ice. However, the general form of the problem has analogies in other systems. The differential equations that describe these systems can be understood to describe a state variable with a certain inertia (imposing a certain relaxation time scale), and processes that can perturb the system away from equilibrium. The concept of stability and the question how slowly a system responds to perturbations can apply to any physical system that can be modelled as a stochastic dynamical system. To illustrate this, we explicitly mention two examples in Sect. 4, namely vegetation dynamics and sea-surface temperatures. Due to the specific focus of our paper on sea ice we refrained from explaining more details here but added references to other studies instead.

5. We now describe the extended RCP8.5 scenario in more detail (see last comment to reviewer 1).

Eisenman, I., and Wettlaufer, J. S.: Nonlinear threshold behavior during the loss of Arctic sea ice, Proc. Natl. Acad. Sci. U. S. A., 106, 28-32, 2009.

Moon, W., and Wettlaufer, J. S.: A low-order theory of Arctic sea ice stability, E. P. L., 96, 39001, 2011.

Moon, W., and Wettlaufer, J. S.: A stochastic perturbation theory for non-autonomous systems, J. Math. Phys., 54, 123303, 2013.

Wagner, T. J. W., and Eisenman, I.: How climate model complexity influences sea ice stability. J. Climate, 28, 3998-4014, 2015a.

Wagner, T. J. W., and Eisenman, I.: Early warning signals for abrupt change raise false alarms during sea ice loss, Geophys. Res. Lett., 10333-10341, 2015b.

Comment by Till Wagner

In agreement with, and in addition to, the insightful and constructive comments of the reviewers, I would like to provide the following feedback to this interesting study, which I think will make a valuable contribution to the literature:

1) Overall, this paper provides interesting and novel insight into the statistical differences between summer and winter sea ice loss, as well as the evolution of sea ice thickness and volume. It therefore goes beyond previous work, including our paper (Wagner & Eisenman, 2015), where we focused on sea ice area during summer. It further fills in important gaps regarding the effects of different types of stochastic forcing.

2) Title and introduction: Regarding the title, I agree broadly with Reviewer 2 that it may be better if the title referred specifically to the statistical indicators, since they are at the core of this study. Regarding the introduction, I would proffer that the focus could be shifted somewhat toward the evolution of variance and autocorrelation under sea ice loss in general, rather than focusing on their (lack of) utility as early warning signals for critical transitions. Introducing the concept of using variance and autocorrelation to help estimate the future mean state and variability of the sea ice cover is an excellent contribution of this study that could be given more weight here in my opinion.

3) Relatedly, a slightly clearer presentation of what has been published on this topic and what is novel, in line with comment 2 by Reviewer 2, may improve the exposition of the paper.

4) As a side note, I want to point out that the spatially explicit model results from Wagner & Eisenman (2015) show an increase in autocorrelation before the loss of the summer sea ice, something that single-column models like E07 may not always pick up on. We suggest that the increase in autocorrelation is due to the growth of the (long-memory) open-water region as the ice retreats, in agreement with the conclusions drawn here.

5) The analysis and discussion of GCM results appears to me (not a GCM expert) very valuable. It highlights a number of important operational limitations in applying statistical indicators as early warning signals, and it provides the first steps toward the use of statistical indicators in GCMs to predict changes in the sea ice cover. I would hope this motivates fruitful further research in the community.

Reference: T.J.W. Wagner and I. Eisenman (2015) "False alarms: How early warning signals falsely predict abrupt sea ice loss", GRL (23) 42, DOI: 10.1002/2015GL066297

We are grateful to Till Wagner for these constructive comments which help to clarify several points in the discussion and will help to improve the manuscript.

1. and 5. We fully agree on these comments concerning what is novel in our study. We emphasise these points in our revised manuscript.

2. We have chosen this general title because our study has relevance beyond the phenomenon of slowing down and early warning signals. What we analyse is the relation between the mean state of Arctic sea ice (or its annual cycle in equilibrium with a certain forcing) and the fast variability around this state. Our main result is that we find a relation between these properties that is fundamental (arising from physical processes) and robust (independent of the model and the description of its variability). Regarding the idea of early warning signals, this is a negative result. Regarding the prospects for stochastic climate models or the inference of past and future climate variability, it is a positive result. Hence, we like to reflect the genericity of our result in the title. We think that this argumentation is in perfect agreement with the rest of the comment, suggesting to focus more on what can be inferred from observations instead of focussing too much on false alarms.

3. We fully agree that we should inform the reader more clearly about the novelty of our manuscript, something we have considered in the revised version.

4. We agree that the inertia of the open ocean causes the increase in autocorrelation in both models. As stated in Wagner and Eisenman (2015b), the autocorrelation of sea-ice volume decreases before Arctic summer sea-ice loss in their model, in agreement with our findings. We note that this happens in all models, also including MPI-ESM which is spatially explicit. As shown in Wagner and Eisenman (2015), there seems to be a somewhat different timing in the onset of slowing down in other variables, like polar temperature and total hemispheric sea-ice area, which tend to increase already before Arctic summer ice is lost. This can occur due to the spatial coupling of grid cells via the atmosphere: As more and more grid cells become ice free with increasing long-wave forcing, the variability of the whole coupled system slows down, which can also affect latitudes where sea-ice is still present, and which can cause a slowing down of the fluctuations of the sea-ice edge's position. For a strict model comparison regarding this issue of the timing, more analysis would be required. We leave this to future studies because it does not affect our results.

Wagner, T. J. W., and Eisenman, I.: Early warning signals for abrupt change raise false alarms during sea ice loss, Geophys. Res. Lett., 10333-10341, 2015b.

---

## Author Response (AR2)

Editor

Dear authors,

 thank you for revising your manuscript according to the suggestions and concerns of the reviewers. I am glad to have received additional comments, which I ask you to carefully consider in your next revisions. Please assure to specifically address comments in report #2 of reviewer #1, and to include a careful revision of the abstract to properly represent the contents and conclusions of the manuscript.

 Thank you and best regards

 Christian Haas

Dear Dr Haas,

we are grateful for these constructive comments that helped to improve the manuscript further. Please find our replies in red font, with quotes from the revised paper in italic font.

On behalf of all authors,

Sebastian Bathiany

Referee 1 (report no. 2)

I previously reviewed this paper, and have positively noted that this version is much improved compared to the earlier version. However, I do still have strong objections to some aspects of the manuscript, which do not allow me to recommend publishing it in this form in the peer reviewed literature. My main objections are related to the motivation of this study, which in my opinion overstates the implications of this study, as the discussion implies there might be bifurcations on the way to an ice-free Arctic, even though it also mentions that comprehensive models do not show that (or only for the opposite state, snowball earth, for paleo simulations, which is a completely different state, and not a state the authors investigate). Similarly, the last sentence of the conclusions again overstates the implications of the study, by suggesting that it can help to provide "early warning for potential extreme events", which isn't supported by the results. Nevertheless, I can see how this study can become publishable after further revisions (and a change of title), limiting its stated implications to what it actually does do. For example, the Abstract reads well now and is limited to what the study does deliver, so I hope the authors can change the title, introduction, and conclusions to match the scope of the Abstract.

We thank the referee for these constructive comments.
We have changed the title, the introduction and the end of the conclusions in the way suggested by the reviewer. We hope that the revised version makes it even clearer that we do not claim the existence of a tipping point in the future. We have also revised the paragraph about the transition to a Snowball Earth state in order to explain its relevance for our study.

 Specific comments:

• I would like to strongly re-iterate that the title should be changed to better reflect the focus of the manuscript on the statistics of sea ice variability on the way to an ice-free Arctic. A changed title was suggested by all reviewers and in the comment by T. Wagner, but it was not changed and no strong reasons were given for why not. I strongly feel that the title is is misleading, and the nature of the paper, focusing on statistical indicators, needs to be reflected in the title, rather than only in the Abstract.

We did not change the title previously because we felt that it captured the recent changes to the paper well (more emphasis on natural variability instead of predicting tipping points); reviewer 2 did in fact not suggest any change to the title. As this seems to be a more important point to the reviewer than to us, we have now nonetheless changed the title completely.

• Line 22: I would suggest rephrasing "To this extent, the prospects to find statistical early warning signals before an abrupt sea-ice loss at a "tipping point" seem very limited." To "Based on these results, the prospects to find statistical early warning signals before an abrupt sea-ice loss at a "tipping point" seem very limited. ", as "to this extent" makes no sense in English.

We thank the reviewer for this helpful comment which we have implemented in the revised manuscript.

• Line 35-37: Isn't one of the main finding that statistical early warning signs are not able to warn about tipping points (as stated in the Abstract)? So this sentence "Moreover, natural climate variability can be an indicator of climate stability and provide "early warning signals" of an approaching tipping point (Scheffer et al., 2009). " needs to be rephrased, maybe to "Furthermore, previous studies (Scheffer et al., 2009) have suggested that natural climate variability can be an indicator of climate stability and provide "early warning signals" of an approaching tipping point.

We agree and have taken up this suggestion in the revised manuscript.

• Page 2, Line 5-9: Tietsche et al. (2012) also clearly showed that sea ice loss is completely reversible, and should be cited. And Wagner and Eisenman (2015) showed why simple models show this bifurcation, and that it was an artifact of the simple models that did not include seasonal cycle or latitudinal differences. So this question has been solved, and should not be presented as a current topic of debate just to make the current study seem more relevant. Comprehensive models don't show bifurcations in sea ice loss, physical understanding does not support it, and the reason simple models nevertheless simulated it has been found. The examples for bifurcations for paleo simulations (which I am not familiar with, but which tend to use models with much reduced resolution compared to present day study) for snowball earth is the opposite of the earlier discussed bifurcation of a irreversible sea ice loss, so it is misleading to cite these here and to imply that therefore there might be bifurcations in the future climate as well. It seems that this only serves to motivate the study presented in this manuscript, but it does not do it in a well reasoned way and this motivation needs to be revised again to be publishable, as it severely currently overstates the implications. The current study has merit as an analysis of statistical indicators of sea ice variability, with some limited implications for the real world, and should be presented as such rather than trying to imply it does more than it can.

We now cite Tietsche et al. (2011) in the introduction, and have again rephrased the paragraph about previous research about potential bifurcations in the future:

"While this "small ice-cap instability" occurred in simplified models (North, 1984; Thorndike, 1992; Eisenman and Wettlaufer, 2009; Abbot et al., 2011), more comprehensive models show a more gradual and reversible sea-ice loss in scenarios of the future (Armour et al., 2011; Tietsche et al., 2011; Boucher et al., 2012; Ridley et al., 2012; Li et al., 2013). Consequently, Wagner and Eisenman (2015a) recently showed in detail how resolving the seasonal cycle and latitudinal differences can eliminate bifurcations in sea-ice models, explaining why oversimplified models lead to wrong conclusions."

We think that this paragraph is in perfect agreement with the reviewer's comment. We then continue by pointing out how complex models differ in the abruptness of sea ice loss:

"Nonetheless, comprehensive models still differ in how abruptly Arctic sea ice area and volume can change (Bathiany et al., 2016). Given the large model uncertainties even in comprehensive models, it is worthwhile to investigate whether changes in certain aspects of the variability are specific to the

*existence of abrupt or even irreversible changes in the future. Observations might then provide an alternative source of information and indicate which model is most reliable in its prediction."*

Undoubtedly, there are many uncertainties in the models, and it is worthwhile to investigate how robust trends in variability are. We do not think (or argue in the paper) that this stands in conflict to the fact that these models all show a reversible sea ice loss.

Concerning the Snowball Earth instability, this effect has indeed been found in state-of-the-art models with a similar resolution than many other studies. We do not argue anywhere that the Snowball Earth instability would also imply the existence of a small ice cap instability and agree with the reviewer that this would be wrong. Our motivation is to study the changes in variability over a vast range of climates from the snowball bifurcation to an ice-free planet, not only future states. Our reference to colder climates than the present is motivated by the fact that our results indicate that one can infer something about sea-ice variability in cold climates as well as hot climates, including the stability of a state that is close to a Snowball bifurcation. To make this clearer, we have rephrased the according paragraph:

*"Interestingly, when cooling the Earth instead of warming it, even comprehensive models show bifurcations, in agreement with simple models (Budyko, 1969; Sellers, 1969). For example, in a complex general-circulation model with current continental distribution and solar insolation, Marotzke and Botzet (2007) identified a globally ice-covered stable state analogous of the 'Snowball Earth' conditions in the Neoproterozoic (Pierrehumbert et al., 2011). Ferreira et al. (2011) and Rose et al. (2013) even found three stable states in a complex model with idealised ocean geometry. Climate variability plays an important role for the likelihood of transitions between such states, and for their reversibility (Lee and North, 1995), and thus needs to be considered to understand the evolution of climate in the Earth's deep past."*

We have also moved an important sentence that explains our motivation from a later point in the introduction to the first paragraph of the introduction:

*"Moreover, understanding the relation between the mean climate and its variability will allow us to draw conclusions about the climate variability in the Earth's deep past, something that is difficult to reconstruct directly (White et al., 2010; Kemp et al., 2015), and that can help to build simple stochastic climate models."*

• Page 13, Line 507: The last sentence of the conclusions is, in my opinion, not supported by the results, and overstates the implications of this study. If the authors are happy to publish this manuscript under a title that clearly reflects that they study "Statistical indicators of sea-ice variability on the way to an ice-free Arctic" (or similar), and remove this final sentence of the conclusions as well as rephrase the introduction, this manuscript should be publishable. But trying to overstate the impactions of the study through the title or the conclusions make it so I can not sign off on for publication in the peer reviewed literature. The Abstract reads fine and does not make this last statement, so it does not seem to be central to the paper.

From the reviewer's general comment we understand that we should not claim to find "early warnings of extreme events" (the penultimate sentence). What we mean here is that our results suggest that a prediction of the variability in general seems possible, not the prediction of specific events and the time of their occurrence. We hope that the following revision of this part makes this clearer:

*"In particular, the strict relation between the mean state of sea ice and its variability suggests the possibility to infer the system's total variability from relatively short observational time series, and to estimate the typical magnitude and longevity of climate anomalies in the future."*

We have now moved the description of our approach to obtain Fig. 4 from Appendix B (which has disappeared) to the main text. Furthermore, we have added several lines that explain certain features of Fig. 4 compared to Fig. 2. The text now reads:

*"In natural systems, the relaxation time usually cannot be measured or calculated as directly as in models. However, one can hope to measure the system's response to natural external perturbations indirectly in form of its variance and autocorrelation. We therefore investigate in stochastic versions of the two column models whether these indicators reflect the changes in timescale. In each experiment, we introduce noise in one of three terms of the equations: To mimic variability in the ocean heat flux (σOHF), we added a Gaussian white noise term to Eq. A1 (Appendix A). To introduce noise to the radiative fluxes, we added the noise term on the radiative balance A (Eq. A2) to perturb the long-wave balance (σLW), or on S (S=1-S_a cos⬚2πt) in Eq. (A2) to perturb the short-wave balance (σSW). We also distinguish small and large noise, as well as white and red noise. In the case of small noise, we choose the noise level in a way that the total variance of E is in the order of 10-9, i.e. much smaller that the amplitude of an annual cycle. In the case of large noise, we adjust the noise level such that the system's stochastic variability is roughly one order of magnitude smaller than the amplitude of the annual cycle, in similarity to the situation in the real world. In case of red noise, we model the external perturbations as an autoregressive process of order one (AR(1) process) with a decorrelation time of 180 days in case of mixed layer energy and 10 days for atmospheric radiation.*

*Fig. 4 shows results for large red noise for all three noise sources. Interestingly, the specific choices for the noise terms hardly affect the results. When introducing small noise to the equations, the evolution of variance and autocorrelation closely follow the results we obtained from the perturbation experiments (Fig. 2a), independent of the noise type. Due to the low temperatures and the large growth-rate of thin ice, the ice coverage A is always close to 1 in winter, and has very small variance regardless of the variability of other variables. In contrast to Fig. 2a, the second peak produced by the ice-albedo feedback is not as pronounced in Fig. 4. This partly results from the lower resolution of the figure (associated with the much larger computational demand), but mostly due to the fact that the natural variability causes the system to cross the tipping point before the deterministic bifurcation point is reached. However, even in case of large red noise, the results are qualitatively similar to Fig. 2a as long as the noise is still small enough to not destroy the whole bifurcation structure of the system. The reason is that the time scales of the variability are still smaller than the typical response time of the ice-mixed layer system. In this regard, the model still sees the imposed noise as white, and the autocorrelation we find is determined by the system's time scale and not the time scale of the red noise. This explains the invariance of the results to the noise type."*

2) While the reviewers have done a much better job at articulating which aspects of the study are new in the main text, this aspect is still lacking in the abstract. Where appropriate, I would have liked to see phrases like 'consistent with previous studies', and 'we here show'.

We have taken up this suggestion. The abstract now reads:

*"We examine the relationship between the mean and the variability of Arctic sea-ice coverage and volume in a large range of climates from globally ice-covered to globally ice-free conditions. Using a hierarchy of two column models and several comprehensive Earth System Models, we consolidate the results of earlier studies and show that mechanisms found in simple models also dominate the interannual variability of Arctic sea-ice in complex models. In contrast to predictions based on very idealised dynamical systems, we find a consistent and robust decrease of variance and autocorrelation of sea-ice volume before summer sea ice is lost. We attribute this to the fact that thinner ice can adjust more quickly to perturbations. Thereafter, the autocorrelation increases, mainly because it becomes dominated by the ocean water's large heat capacity when the ice-free season becomes longer. We show that these changes are robust to the nature and origin of climate variability in the models and do not depend on whether Arctic sea-ice loss occurs abruptly or irreversibly. We also show that our climate is changing too rapidly to detect reliable changes in autocorrelation of annual time series. Based on these results, the prospects of detecting statistical early warning signals before an abrupt sea-ice loss at a "tipping point" seem very limited. However, the robust relation between state and variability can be useful to build simple stochastic climate models, and to make inferences about past and future sea-ice variability from only short observations or reconstructions."*

3) I particularly liked the results presented in Fig. 7, suggesting that at the current rate our climate is changing too rapidly to detect significant changes in variance. I would recommend including a sentence on that result in the abstract.

We have taken up this suggestion (see above).

[revised manuscript text omitted]